# Exploring risk and protective factors which distinguish suicidal and self-harm behaviours from suicidal and self-harm ideation in young people: A systematic review

**Marianne E. Etherson**[1]\*, **Sieun Lee**[2,3], **Krystyna J. Loney**[1], **Isobel P. Steward**[2], **Joey Ward**[1], **Heather McClelland**[1], **Aaron Kandola**[4], **Josimar A. De Alcantara Mendes**[5], **Chris Hollis**[2,6], **Ellen Townsend**[7], **Dorothee P. Auer**[2,3], **Rory C. O'Connor**[1] on behalf of the Digital Youth Research Programme

**1** Suicidal Behaviour Research Lab, School of Health and Wellbeing, University of Glasgow, Glasgow, United Kingdom, **2** Mental Health & Clinical Neurosciences, School of Medicine, University of Nottingham, Nottingham, United Kingdom, **3** Precision Imaging Beacon, University of Nottingham, Nottingham, United Kingdom, **4** MRC Unit of Lifelong Health and Ageing, University College London, London, United Kingdom, **5** Responsible Tehcnology Institute, University of Oxford, Oxford, United Kingdom, **6** National Institute of Health Research (NIHR) MindTech HealthTech Research Centre, Institute of Mental Health, School of Medicine, University of Nottingham, Nottingham, United Kingdom, **7** School of Psychology, University of Nottingham, Nottingham, United Kingdom

\* Marianne.Etherson@glasgow.ac.uk

## Abstract

### Background

Self-harm and suicidal thoughts and behaviours among young people are significant global public health concerns. Although most young people with thoughts of self-harm or suicide do not act on their thoughts, it is important to identify factors that distinguish thoughts of self-harm and suicide from behaviours. To date, there are no reviews distinguishing self-harm and suicidal behaviours from thoughts of self-harm and suicide in young people or that have synthesised factors distinguishing self-harm behaviours from self-harm ideation. The current review addresses these gaps in the literature.

### Methods

We systematically searched: CINAHL, Embase, Medline, PsycINFO, Psychology and Behavioural Sciences Collection, and Web of Science Core Collection for articles published between 2011 and April 2024. Ninety-nine studies met inclusion criteria, with 92 articles examining risk and protective factors that distinguished suicide attempts from suicidal ideation and seven articles examining factors that distinguished self-harm behaviours from self-harm ideation. Using a narrative synthesis approach, studies were grouped by their outcome variable (e.g., self-harm or suicide) and then by risk and protective factors.

**Data availability statement:** All relevant data are within the paper and its Supporting Information files.

**Funding:** The authors acknowledge the support of the UK Research and Innovation (UKRI) Digital Youth Programme award which is part of the AHRC/ESRC/MRC Adolescence, Mental Health and the Developing Mind programme. Grant number: MR/W002450/1 The funders had no role in study design, data collection and analysis, decision to publish, or preparation of the manuscript.

**Competing interests:** The authors have declared that no competing interests exist.

## Results

While findings were inconsistent, the presence of non-suicidal self-injury, physical, emotional, or sexual abuse, violence, and family factors (e.g., family conflict) distinguished suicidal attempts from suicidal ideation. By contrast, the presence of parental factors (e.g., parental connectedness) and greater academic achievement were protective and distinguished suicidal ideation from suicide attempts. Being female, exposure to self-harm/suicide, and impulsivity distinguished self-harm behaviours from self-harm ideation. There was no evidence of protective factors that distinguished self-harm behaviours from self-harm ideation.

## Conclusions

The current review highlights important risk and protective factors that distinguish suicidal and self-harm behaviours from suicidal and self-harm ideation in young people. Our review has important implications for intervention and prevention efforts as identifying key risk and protective factors can improve risk assessment for young people experiencing thoughts of self-harm and suicide and enable more targeted interventions.

## Introduction

Self-harm and suicidal behaviours in young people are significant public health concerns. Despite challenges in defining self-harm and suicide attempts, self-harm can be broadly defined as self-injurious behaviour irrespective of motive; self-harm is a broad term that can include non-suicidal self-injury (self-injurious behaviour without intent to die) [1,2]. By contrast, self-harm ideation depicts thoughts of self-harm without self-injurious behaviour [2]. Suicide attempts are defined as self-injurious, non-fatal behaviours with at least some evidence of intent to die [1]. Whereas, suicidal ideation is defined as thoughts of ending one's life, where the risk of suicide can progress from relatively passive thoughts to more active thoughts [1]. Globally, suicide is the third leading cause of death among 15- to 29-year-olds [3]. In addition, a recent meta-analysis examining the global prevalence of self-harm among non-clinical adolescents, found that 22% of adolescents engaged in self-harm behaviours at least once in their lifetime [4]. Evidence also suggests that self-harm and suicidal behaviours are on the rise in young people [5,6]. While many young people experience thoughts of self-harm and suicide, approximately only a third act on their thoughts [7]. It is, therefore, important to better understand the risk and protective factors associated with whether young people with thoughts of self-harm or suicide will engage in self-harm or suicidal behaviours.

Suicide is complex and manifests from an interplay of biological, psychological, social, existential, and cultural factors [8]. Though our understanding of the risk factors for suicide has grown exponentially in recent decades [8], our ability to predict suicide is no better than it was 50 years ago [9]. One possible reason for this it that

many common risk factors for suicide did not distinguish between risk factors for suicidal ideation and risk factors involved in the transition from suicidal ideation to suicide attempts [10]. For instance, evidence suggests that many previously identified risk factors for suicide (e.g., depression, hopelessness) are risk factors for suicidal ideation, but not attempts and/or do not significantly distinguish between those with thoughts of suicide and those who attempt suicide [11–13]. The distinction between the development of suicidal ideation and the transition from suicidal ideation to suicide attempts is now widely accepted among researchers and is incorporated into current theoretical models of suicidal behaviour [14,15].

Building on earlier models, Joiner [16] made a critical advance in the literature by proposing that suicidal ideation and attempts have their own distinct risk factors. Since the emergence of Joiner's Interpersonal Psychological Theory of Suicide [IPTS [16,17];], more contemporary theories of suicide, such as the Integrated Motivational-Volitional Model of suicidal behaviour [IMV [18,19];] and the Three Step Theory [20], have adopted an "ideation-to-action framework" (see Table 1). This framework acknowledges that suicidal thoughts and suicide attempts are separate processes with distinct risk factors [14]. Research has largely supported these theoretical models and the role of acquired capability, volitional factors, and contributors to the capacity for suicide involved in the transition from ideation-to-action [21–23]. Notably, while the "ideation-to-action" framework was developed to better understand suicidal behaviour, it can also extend to self-harm behaviours [ [24]] and is instrumental to better understand those at risk of both self-harm and suicide.

Two existing reviews have examined the risk factors that distinguish suicide attempts from suicidal ideation through an ideation-to-action framework [10,25]. In the earlier review, May and Klonsky [10] meta-analysed 27 cross-sectional studies comparing risk factors for suicide attempts versus suicidal ideation. Anxiety disorders, post-traumatic stress disorder, drug use disorders and sexual abuse history were the only risk factors to distinguish suicide attempts from suicidal ideation. More recently, Bayliss et al. [25] conducted a systematic scoping review of suicide capability (i.e., a factor by which an individual feels capable of making a suicide attempt). Factors which distinguished suicide attempts from suicidal ideation included painful and provocative events (e.g., abuse, maltreatment), genetic polymorphisms, interoceptive deficits (i.e., decreased ability to perceive bodily sensations), neuroticism, and impaired cognitive functioning.

Although these reviews have been valuable in advancing our understanding of suicide enaction, they are limited to adult populations only, while no reviews have focused on young people [10,25]. Bayliss et al. [25] suggested there may be unique factors involved in the transition from ideation to action in young people (e.g., decision-making capabilities).

**Table 1. An overview of the main theoretical models based on the 'ideation-to-action' framework.**

| 'Ideation-to-action' models of suicidal behaviour | Overview of the model |
|---|---|
| The Interpersonal Psychological Theory of Suicide (IPTS [16,17];) | • The IPTS suggests that individuals must possess the desire and capability to die by suicide.<br>• Both perceived burdensomeness and thwarted belongingness are necessary to produce suicidal desire.<br>• However, individuals must also acquire capability for suicide (fearlessness of death and greater tolerance of physical pain) to die from suicide. |
| The Integrated Motivational-Volitional Model of Suicidal Behaviour (IMV [18,19];) | • The IMV model is a tripartite diathesis-stress model which maps out background factors and triggering events (*pre-motivational phase*), the formation of suicidal ideation (*motivational phase*) and the transition from suicidal ideation to behaviour (*volitional phase*).<br>• The *pre-motivational phase* outlines a broader context in which suicide may occur.<br>• In the *motivational phase*, defeat leads to entrapment in the presence of threat-to-self moderators (e.g., problem solving) and entrapment leads to suicidal ideation in the presence of motivational moderators (e.g., thwarted belongingness).<br>• In the *volitional phase*, the presence of volitional moderators (e.g., access to means, exposure to suicide) increases the likelihood of acting on suicidal ideation. |
| The Three-Step Theory [20] | • The *first step* of this model proposes that the interaction of pain and hopelessness leads to suicidal ideation.<br>• The *second step* suggests that suicidal ideation escalates when pain exceeds connectedness.<br>• The *final step* suggests that suicidal ideation progresses to suicide attempts only when one has the capacity to attempt suicide. |

In addition, because adolescence is a period defined by increased interpersonal sensitivity, impulsivity, and risk-taking behaviours [26], young people are likely to differ from adults in their motivations for attempting suicide. It is possible, therefore, that previously identified risk or protective factors implicated in the transition from ideation to attempts in adults, may not be relevant in young people and there may be unique factors which are specific to younger populations. Consequently, a review identifying the factors involved in this transition in young people is warranted.

Despite research extending the ideation-to-action framework to self-harm [24], to date, no reviews have synthesised the distinct risk or protective factors that distinguish self-harm behaviours from self-harm ideation. Self-harm behaviours are one of the strongest predictors of suicide attempts and are prevalent in young people [27], therefore it is important to synthesize evidence of the factors that may be involved in the transition from self-harm ideation to self-harm behaviour in young people. Thus far, no reviews have synthesised existing work on the distinct risk and protective factors that distinguish young people who have thoughts of self-harm or suicide from those who act on their thoughts. The current review addresses these existing limitations by being the first to synthesise the distinct risk or protective factors which distinguish suicide attempts from suicidal ideation and distinguish self-harm behaviours from self-harm ideation in young people. This comprehensive review provides an important touchstone, synthesising the most up-to-date evidence on the ideation-to-framework in young people, which can be utilised to guide future research, highlight knowledge gaps, and inform future preventative and treatment efforts.

### Research question

What risk and protective factors distinguish young people (aged 13–25 years) who have thoughts of self-harm or suicide from those who act on them?

## Methods

### Search strategy

The current review was registered on PROSPERO (CRD 42022332224) where the search strategy was predefined. The search strategy followed Preferred Reporting Items for Systematic Reviews and Meta-Analyses (PRISMA) guidelines [28]. The following databases were systematically searched: CINAHL, Embase, Medline, PsycINFO, Psychology and Behavioural Sciences Collection, and Web of Science Core Collection. The original search was conducted on 3rd May 2022 and updated on 11th April 2024. The search was restricted to English language, peer-reviewed articles, and articles published between 2011–2022. The start date of 2011 was chosen based on the publication of the updated IPTS [17] and emergence of the IMV model [18], both ideation-to-action frameworks. Separate searches were conducted independently for each database using the following search terms and Boolean phrases: (i) Thought* OR thinking OR ideat* AND behavi* OR enact* OR attempt*, (ii) self-harm* OR self-injur* OR self-mutilat* OR suicid* OR ideation-to-action, (iii) young people OR youth OR adolescen* OR young adult* OR teen*. The final search combined searches 1–3.

### Inclusion and exclusion criteria

For inclusion in the review, studies had to: (a) be a peer-reviewed primary empirical paper of any study type (e.g., quantitative, qualitative, mixed methods); (b) be available in English language (c) include a sample of young people aged between 13–25 years or include a sample with a broader age range than 13–25 years, but where the $M_{age}$ of participants fell into this range; (d) examine risk or protective factors that distinguished suicide attempts from suicidal ideation or self-harm behaviour from self-harm ideation; and (e) include at least two distinct groups (e.g., a suicidal ideation group with no previous history of suicide attempts vs a suicide attempt group or a self-harm ideation group with no previous history of self-harm behaviour vs a self-harm group) or track the transition from suicidal ideation at baseline (with no prior history of suicide attempts) to suicide attempts or self-harm ideation at baseline (with no prior history of self-harm) to self-harm behaviour over time. While there is no universally established age range for young people, based on various definitions

and because suicide attempts are rare below age 13 years [29], and as adolescence is thought to extend until the mid-twenties [30], our classification of young people ranged from 13–25 years.

Conversely, studies were excluded if they: (a) were not a peer-reviewed primary empirical paper (e.g., a review article, case study or commentary); (b) were not available in English language (c) did not include a sample of young people within the age range of 13–25 years or where the majority of participants fall into this range; (d) did not examine risk or protective factors that distinguished suicide attempts from suicidal ideation or self-harm behaviour from self-harm ideation and (e) did not examine two distinct groups (e.g., a suicidal ideation group with no previous history of suicide attempts vs a suicide attempt group or a self-harm ideation group with no previous history of self-harm behaviour vs a self-harm group) or track the transition from ideation to behaviours over time.

## Screening

Search results from each database were exported to Endnote. Duplicates were removed electronically in Endnote and manually when articles were exported into an Excel document. The first author assessed the retrieved studies' titles and abstracts for inclusion. In cases where it was unclear if a study met the inclusion criteria, the study was retained to be screened by its full text. To check for accuracy of the screening process, a second assessor screened a random 10% sample of title and abstracts and a random 20% sample of full-text papers. The two reviewers reached excellent agreement for title and abstract screening (Kappa (K) = 0.81) and substantial agreement for full-text screening (K = 0.65). Any disagreements were resolved through discussion.

## Data extraction

Data were extracted by the first author and 10% were cross-checked by a member of the research team. The first author contacted authors of included studies to request any relevant data not reported within manuscripts or supplementary material. The data extracted included author and date of publication, sample type and demographics, study design, country, risk and protective factors examined, outcome measure, significant risk or protective factors found and study results.

## Quality assessment

The final studies were assessed for quality. Each study was assigned an overall methodological quality score. The characteristics assessed were guided by a methodological quality assessment instrument, tailored for the current review (see S1 Table for a quality assessment tool and scoring guide). The methodological characteristics evaluated included: (1) methodological design, (2) power calculation of sample size, (3) assessment of suicide and self-harm measures (4) assessment of risk/protective factor measures and (5) confounding variables controlled for. Quality assessments were classified as: 0–2 = very low quality, 3–4 = low quality, 5–6 = reasonable quality, 7–8 = good quality, and = 9–12 excellent/very good quality. Higher scores indicated a lower likelihood of methodological bias and were used to aid interpretation of the study findings. The first author conducted the quality assessment, and a member of the research team assessed 20% of the methodological quality scores for inter-rater reliability. Any disagreements in scores were resolved via discussion with 100% post-discussion concordance. When comparing studies rated low quality (≤ 2; e.g. [31–33],) vs excellent/ very good quality (≥ 9; e.g. [34–36],), there was no evidence of an association between study quality and the nature or significance of the findings.

## Data analysis and synthesis

A narrative synthesis was conducted, based on guidelines for reporting a systematic review synthesis, without meta-analysis [37]. Studies were grouped by their outcome variable (e.g., self-harm or suicide) and then by risk and protective factors.

## Results

A total of 22,518 articles were retrieved from the initial search and the updated search. Following deduplication, there were 7,070 articles. After applying eligibility criteria at the title and abstract screening stage, 1,176 articles remained. The full text of 1,176 articles were then screened. The main reasons for exclusion included: not being a primary empirical paper (e.g., narrative reviews, systematic reviews, meta-analyses, commentaries), participants not being in the required age range (based on the $M_{age}$), examining self-harm/suicidal ideation only or self-harm/suicidal behaviours only, not examining risk or protective factors for self-harm and suicidal ideation and behaviours, and examining ideation and behaviours together as one outcome (e.g., suicidality). Most excluded studies examined risk or protective factors for self-harm ideation and self-harm behaviour or suicidal ideation and suicide attempts, but did not examine risk or protective factors distinguishing between ideation and behaviours. Studies that compared a suicidal ideation group to a control group or a suicide attempt group to a control group, were also excluded. In addition, in line with May and Klonsky's [10] meta-analytical review, studies examining factors that distinguished between a self-harm ideation group and a self-harm behaviour group or a suicidal ideation group and a suicide attempt group, but did not include a true ideation group (i.e., a proportion of the ideation group had previous history of self-harm or suicidal behaviour, respectively) were excluded.

After screening the full text of 1,176 articles, 96 studies met the inclusion criteria. An additional three articles were identified through handsearching. Therefore, 99 articles were included in this review, with 92 articles examining factors that distinguished suicide attempts from suicidal ideation, and seven articles examining factors that distinguished self-harm behaviour from self-harm ideation (see Fig 1; see Table 2 and 3). All studies that met the inclusion criteria and thus included in the review were quantitative.

## Narrative synthesis

### Study characteristics of studies examining risk and protective factors distinguishing suicide attempts from suicidal ideation

Ninety-two studies examined factors that distinguished suicide attempts from suicidal ideation. The sample size of included studies ranged from 41 to 73,238. Most studies included both female and male samples alongside a small percentage of gender minorities in some studies, whereas two studies were in female only samples [63,124]. The included studies were conducted in the United States (US; $N=52$), China ($N=9$), Australia ($N=6$), United Kingdom ($N=5$), Canada ($N=3$), Italy ($N=3$), Korea ($N=3$), Germany ($N=2$), Portugal ($N=1$), New Zealand ($N=1$), Turkey ($N=1$), Ethiopia ($N=1$), Lebanon ($N=1$), Iran ($N=1$), Thailand ($N=1$) and Brazil ($N=1$). One study was conducted across eight countries (Australia, Belgium, Germany, Mexico, Northern Ireland, South Africa, Spain, and the US). All studies were quantitative; seventy-five studies were cross-sectional, 16 were longitudinal and one was experimental [34]. No qualitative studies met the inclusion criteria. Fifty-four studies utilised validated multi-item measures of suicidal ideation and attempts, 36 studies utilised non-validated scales or single item questions. Two studies did not include how suicide ideation/ behaviour was assessed. Most suicide measures were self-report, however one study identified suicidal ideation and attempts through a retrospective analysis of medical records [35].

### Methodological quality

Quality assessment criteria and scores for studies examining suicidal ideation and behaviour are reported in the supplementary material (see S2 Table). The maximum score obtainable was 12. The mean score across studies was $5.80\pm2.03$ (range 1–10). The lowest scoring domain was power calculation.

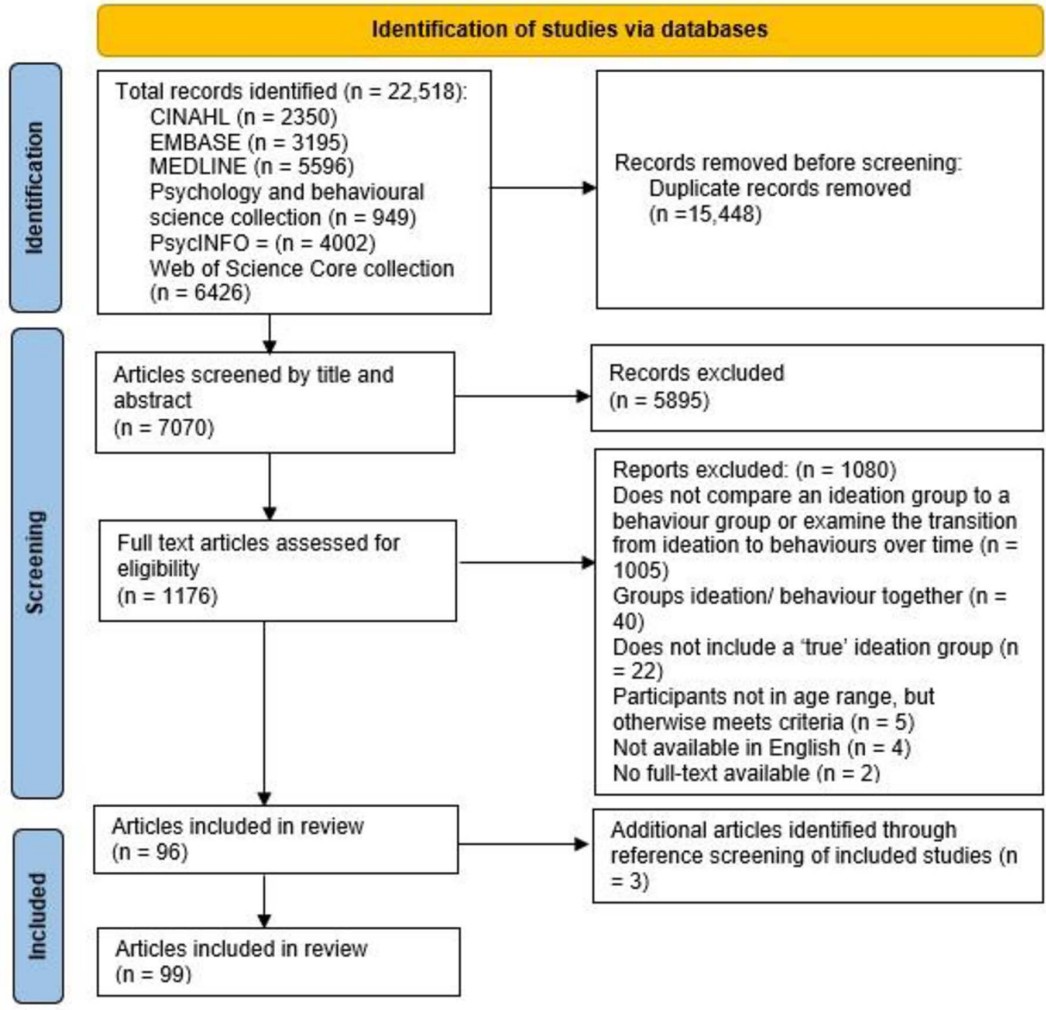

**Fig 1. PRISMA diagram.**

## Risk factors distinguishing suicide attempts from suicidal ideation

### Sex/ gender

Twenty-nine studies examined gender. Sixteen of these found that suicide attempt groups had a significantly greater number of females, relative to suicidal ideation groups (e.g. [59,66,68],). Whereas, thirteen studies found that gender did not significantly distinguish suicide attempts from suicidal ideation (e.g. [44,69,75],). Conversely, one study found male gender to significantly distinguish suicide attempts from suicidal ideation [144].

### Non-suicidal self-injury (NSSI)

Fifteen studies examined NSSI. Of these, 12 identified NSSI to be a risk factor that distinguished suicide attempts from suicidal ideation (e.g. [69,71,113],). Three cross-sectional studies did not find NSSI to significantly distinguish suicide attempt groups from suicidal ideation groups (e.g. [68,83,96],). In addition, two studies reported that a greater number of NSSI methods was also a risk factor (e.g. [12,137],).

**Table 2. Study summary table for suicidal ideation and attempts.**

| Author and date | Sample type and demographics | Study design | Country | Exposure: risk or protective factors measured | Suicide measure | Significant risk or protective factors for suicide attempts | Study findings |
|---|---|---|---|---|---|---|---|
| Ahmad-boukani et al [38]. | 600 participants, (76.8% female), $M_{age}$ =23.01 years, $SD$=4.32. | Cross-sectional | Iran | Fearlessness about death, and suicide capacity (acquired, dispositional, and practical capacity). | SBQ-R [39] SA: "How many times have you attempted to commit suicide in the past when you to some extent intended to die?" | Practical capacity. | For suicidal capacity, only practical capacity ($t$=2.09; $p$<0.05) differentiated the SA group from the SI group. However, fearlessness about death ($t$=0.921; $p$=0.359) and acquired capacity ($t$=0.02; $p$=842) had no significant effect. |
| Ai et al [40]. | 80 young adults with SI (56.25% female), $M_{age}$ =19.96 years, $SD$=1.36. Sixteen participants had past suicide attempts (50% female), $M_{age}$ =20.56, $SD$=1.03. | Cross-sectional | China | Suicidal ideation and intent. | SI: BSS (measures current suicidal ideation) [41]. For those with SI, a question on SA was asked to assess whether they had ever attempted suicide. | SI and suicidal intent. | The SA group were significantly different from the SI group in suicidal intention scores ($t$=5.53, $p$<0.05). |
| Alarcón et al [42]. | 120 depressed adolescent inpatients aged 11–18 years. SA group: $N$=24, 58% female, $M_{age}$ =14.61 years, $SD$=1.57. High SI group: $N$=27, 56% female, $M_{age}$ =15.04 years, $SD$=1.68. Low SI group: $N$=31, 55% female, $M_{age}$ =14.87 years, $SD$=1.75. Control group: $N$=38, 50% female, $M_{age}$ =14.46 years, $SD$=1.52. | Cross-sectional | US | Depression severity, depression diagnoses, level of SI, head motion, amygdala connectivity, amygdala functioning during an emotional self-face recognition task. | K-SADS-PL [43] | Depression severity (compared to low SI but not high SI), MDD diagnosis. Greater SI (compared to low SI but not high SI), lower prevalence of depressive disorders not otherwise specified. Greater amygdala connectivity with the dlPFC/ dACC, dmPFC, and precuneus/ cuneus (compared to low SI, but not high SI), and greater left amygdala rACC. | The SA and high SI groups had greater depression severity and were more likely to be diagnosed compared to the low SI group. The SA and high SI group also endorsed higher SI, compared to the low SI group. The SA group had lower prevalence of a diagnosis of depressive disorder not otherwise specified, compared to the high SI and low SI groups. SA and high SI showed greater amygdala connectivity with dlPFC/dACC (High SI: $t$691=3.92; SA: $t$691=3.45), dmPFC (High SI: $t$691=4.08; SA: $t$691=3.57), and precuneus/cuneus (High SI: $t$691=3.60; SA: $t$691=3.18), compared to the low SI group. Moreover, the SA group showed greater left amygdala-rACC connectivity, compared to both SI groups (SA> high SI: $t$289=3.05; SA> low SI: $t$313=2.93). |

(Continued)

**Table 2.** (Continued)

| Author and date | Sample type and demographics | Study design | Country | Exposure: risk or protective factors measured | Suicide measure | Significant risk or protective factors for suicide attempts | Study findings |
|---|---|---|---|---|---|---|---|
| Alqueza et al [44]. | 970 adolescents (71.06% female), 12–19 years, $M_{age}$ =15.58, $SD$=1.42, 81.09% White, 2% Black, 5.57% Asian, 0.32% Pacific Islander, 0.74% Native American, 10.29% multicultural. | Retrospective, Cross-sectional | US | Age, sex, lifetime depressive disorder, current anxiety disorder, number of psychiatric disorders, lifetime physical abuse, lifetime sexual abuse, NSSI, knowing a peer who had made a suicide attempt. | SITBI [45] Measured lifetime and 12-month prevalence and frequency of suicidal ideation, plans, and attempts. | Lifetime physical abuse (with and without a suicide plan); lifetime NSSI injury (without a plan); and knowing a peer who had made a suicide attempt (with a plan). | Only lifetime physical abuse among the SI group without a plan (aOR = 2.67, 95% CI 1.31–5.42), and with a plan (aOR = 1.82, 95% CI 1.13–2.95), lifetime NSSI injury among the SI group without a plan (aOR = 2.11, 95% CI 1.06–4.18), and knowing a peer who had made a SA among the SI group with a plan (aOR = 1.93, 95% CI 1.33–2.80) significantly distinguished the SA group from the SI group. |
| Auerbach et al [46]. | 101 adolescent inpatients (81.9% female), 13–19 years $M_{age}$=15.50, $SD$=1.48, 82.70% White, 9.2% multicultural, 7.1% Asian, and 1.0% Black or African American. SI group: 72.7% female, $M_{age}$=15.78 years, $SD$=1.52. SA group: 91.3% female, $M_{age}$=15.15 years, $SD$=1.37. | Cross-sectional | US | Anhedonia, SI, depressive symptoms, anxious symptoms, abnormal effort-cost computations. | Beck Scale for Suicide Ideation (SSI; 19 items measure current suicidal ideation [41]). Self-report and interview. SA was reported within the past year. | Female, greater anhedonia severity; abnormal/ low effort-cost computations (Those who attempted suicide were less likely to pursue high value rewards when outcomes are uncertain). | The SA group were significantly more likely to be female, relative to the SI group ($\chi$[1, $N$=101] = 5.17, $p$=0.017)). The SA group reported significantly greater levels of anhedonia compared to the SI group, when controlling for depressive symptoms, anxious symptoms, and SI ($F_{1,96}$=4.05, $p$=0.047, $\eta^2$=0.04). The SA group reported significantly lower high effort-cost computations than the SI group ($F_{1,91}$=7.51, $p$=0.007; $\eta^2$=0.08). There were no significant differences between the SA and SI group on SI, depressive symptoms, or anxious symptoms. |
| Berny & Tanner-Smith [47] | 287 adolescents discharged from inpatient or outpatient substance use disorder treatment (44.9% female), 13–19 years, $M_{age}$ =16.30, $SD$=1.1, 72.50% White. | Cross-sectional | US | Weapon violence victimization, physical abuse, sexual abuse, internalizing disorder severity (i.e., MDD, GAD), substance abuse (past 3-month use of alcohol, cannabis, and illicit drugs). | M.I.N.I. Structured Clinical Interview for DSM-IV (M.I.N.I.-SCID [48]). SI: "In your lifetime, was there a time when you thought you would be better off dead or wish you were dead?", "During that time when you felt that you'd be better off dead, did you think about suicide?". SA: "Did you attempt suicide?". | Weapon violence victimization; physical abuse; sexual abuse; alcohol use; interaction of MDD severity and frequency of alcohol use. | The SA group reported significantly greater prior weapon violence victimization (OR = 2.21), physical abuse (OR = 2.66), and sexual abuse (OR = 3.38) than the SI group. The interaction of depressive disorder severity and frequency of alcohol use predicted suicide attempts ($b$=0.01*, RRR=1.01, 95% CI 1.00–1.03). |

*(Continued)*

**Table 2.** (Continued)

| Author and date | Sample type and demographics | Study design | Country | Exposure: risk or protective factors measured | Suicide measure | Significant risk or protective factors for suicide attempts | Study findings |
|---|---|---|---|---|---|---|---|
| Berona et al [49]. | 1006 sexual and gender minority youths (29.74% female), $M_{age}$ =20.37 years, $SD$=3.2, 24% White, 35% Black, 29% Latinx, 12% other. | Longitudinal. Follow-ups conducted over 16.56 ($SD$=3.7 months) range 6–24 months). | US | Social support, Computerized Adaptive Test for Suicide (CAT-SS), Computerized Adaptive Diagnostic Test for Major Depressive Disorder (CAD-MDD), Computerized Adaptive Test–Depression Inventory (CAT-DI), alcohol use, depressive symptoms, sexual and gender minority youth victimization. | Single item measures from the Youth Behavior Survey. SI: "During the past 6 months, did you seriously consider attempting suicide". SA: "During the past 6 months, how many times did you actually attempt suicide". | Social support, CAT-SS, sexual and gender minority youth victimization. | Among youths with a history of SI, social support reduced the risk for attempts (HR=0.66, 95% CI 0.45–0.96). Among youths with a history of SI, predictors of the transition from SI to SA included baseline CAT-SS score (HR=1.51, 95% CI 1.06–2.15) and victimization (HR=2.48, 95% CI 1.10–5.59). |
| Burke et al [50]. | 520 undergraduate students. SI group: $N$=329 (75.8% female), $M_{age}$=20.53 years, $SD$=2.67). Suicide plan group: $N$=125 (71.3% female), $M_{age}$=20.80 years, $SD$=3.35. SA without intent to die group: $N$=36 (91.7% female), $M_{age}$=20.78 years, $SD$=2.60. SA with intent to die group: $N$=30 (79.3% female), $M_{age}$=20.60 years, $SD$=1.94. | Cross-sectional | US | Physical and emotional abuse, physical and emotional neglect, alcohol use, drug use, eating restraint behaviour, physical aggression, NSSI lifetime frequency, acquired capability. | One item from the SBQ-R: "Have you ever thought about or attempted to kill yourself" [39]. | Emotional abuse, physical abuse, childhood physical neglect, NSSI lifetime frequency. | Those who attempted suicide with intent to die reported higher levels of emotional abuse ($p<.001$, $d$=1.19), physical abuse ($p<0.001$, $d$=0.67), physical neglect ($p$=0.003, $d$=0.53), than those reporting SI only. Those who attempted suicide reported significantly greater NSSI lifetime frequency ($p<0.001$). There were no significant differences between groups on alcohol use ($p$=0.28), drug use ($p$=0.01.), eating restraint ($p$=0.28), childhood emotional neglect ($p$=0.06), childhood sexual abuse ($p$=0.46), acquired capability ($p$=0.08), and physical aggression ($p$=0.02). |

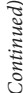

*(Continued)*

| Author and date | Sample type and demographics | Study design | Country | Exposure: risk or protective factors measured | Suicide measure | Significant risk or protective factors for suicide attempts | Study findings |
|---|---|---|---|---|---|---|---|
| Calear et al [51]. | 1382 non-clinical community-based adolescents (57% female), 12–17 years, $M_{age}$ =13.3, $SD$=1.2. Adolescents reporting SI: $N$=167. | Cross-sectional | Australia | Perceived burdensomeness, thwarted belongingness, capability for suicide and interactions of the above variables. | Single item measures from the Youth Risk Behaviour Survey. SI: "Seriously consider killing yourself". SA: Report the number of times they had killed themselves in the past 12 months. Responses were dichotomised into (0) no attempts and (1) one or more attempts. | Two-way interaction between thwarted belongingness and capability for suicide. | The two-way interaction between thwarted belongingness and capability for suicide significantly predicted SA among those reporting SI ($p$=0.018). In the final model with age, gender, and depression included, the two-way interaction remained significant ($p$=0.029). |
| Castaldo et al [52]. | 294 adolescent emergency department patients admitted for SI or SA (69.39% female), 11–17 years, $M_{age}$ =15.5 years, $SD$=1.8. Ethnicity not reported. | Cross-sectional/ retrospective | Italy | Psychiatric diagnoses, substance abuse. | Italian version of the C-SSRS [53]; K-SADS-PL [43] measured current SA and SI. | Substance abuse | Those who were hospitalised with a SA reported significantly higher levels of substance abuse ($c^2$=4.2, $p$=0.04), than those hospitalised with SI. |
| Chu et al [34]. | 107 undergraduate students (65.2% female), 18–35 years, $M_{age}$ =19.3 years, $SD$=2.5, 74.8% White, 7% Black, 5.2% Asian, 0.9% American Indian/ Alaskan Native, 4.3% other, 7.8% not reported. SI group: ($N$=14) $M_{age}$ =19.5 years. SA group: ($N$=17) $M_{age}$ =20.2 years. | Experimental | US | SI, immediate to delayed task recall field and observer perspectives, mood ratings, vividness of visual imagery ability. | Revised Suicide Attempt and Self-Injury Interview (SASII [54]:) measures first, most recent, and/or most medically severe suicide attempts. DSI-SS [55] measured current suicidal ideation. | Greater SI | The SA group reported significantly greater SI ($β$=.18, $p$=.05, $pt^2$=.08), compared to the SI group, even after controlling for depressive symptoms ($β$=.20, $p$=.004, $pt^2$=.09). For immediate event recall, the SA group's recall of a previous SA did not significantly differ from the SI group's recalls of a negative event with regard to the field ($β$=−.09) and observer ($β$=.03) perspectives, controlling for time since the event. During recall of a neutral event, the SA group recalled the events more from the field ($β$=.24, $p$=.017, $pt^2$=.06) and less from the observer ($β$=−.22, $p$=.027, $pt^2$=.05) perspective compared to the SI group. For immediate event recall, the SA group's recall of a previous SA did not significantly differ from the SI group's recalls of a negative event with regard to the field ($β$=−.09) and observer ($β$=.03) perspectives, controlling for time since the event. There were no significant differences in mood ratings and vividness of visual imagery ability between groups. |

*(Continued)*

| Author and date | Sample type and demographics | Study design | Country | Exposure: risk or protective factors measured | Suicide measure | Significant risk or protective factors for suicide attempts | Study findings |
|---|---|---|---|---|---|---|---|
| Claudius & Axeen [35] | 169,047 young patients visiting the emergency department, aged 5–19 years. SI group: (58% female) $M_{age}$ =15.3 years, $SD$ =not reported. SA group: (71% female), $M_{age}$ =15.7 years, $SD$ =not reported. | Retrospective, longitudinal | US | Concurrent anxiety, personality disorder, alcohol-related disorder, psychotic disorder, ADHD, substance related disorder, mood disorder. | SI was identified by ICD-9 code V62.84 and SA was identified by ICD-9 codes E950.x to E959.x. | Greater anxiety disorders, personality disorders, and alcohol-related disorders, lower comorbid psychosis diagnosis, ADHD, and substance-related disorder. | The SA group reported significantly greater anxiety disorders (−0.089, 95% CI −0.094, −0.084), personality disorders (−0.009, 95% CI −0.012, −0.007), and alcohol-related disorders (−0.007, 95% CI −0.009, −0.004) than the SI group. The SA group reported significantly lower comorbid psychotic disorder (0.021, 95% CI 0.018, 0.023), ADHD (0.045, 95% CI 0.041, 0.049), and a substance-related disorder (0.021, 95% CI 0.017, 0.025) compared to the SI group |
| Commisso et al [56]. | 2143 youth, approx. 15 and 22 years (when SI/SA were measured). Sex/ gender, $M_{age}$, $SD$, and ethnicity not reported. | Longitudinal | Canada | Externalizing, internalizing, and comorbid problems. | DISC (self/ parental-report) and the Diagnostic Interview Schedule for Adults (self-report). SI: "Have you/ your child thought seriously about killing yourself/ themselves?". SA: "Have you/your child tried to kill yourself/ themselves?'. Administered at age 15 and 22 years. | Externalizing and comorbid problems (measured from age 6–12 years). | The SA group reported significantly greater externalizing (aOR = 2.01, 95% CI 1.29–3.12) and comorbid problems (aOR = 2.28, 95% CI 1.29–4.03) than the SI group. There were no significant differences in internalizing problems between groups (aOR = 1.67 95% CI 0.99–2.81). |
| Cramer et al [57]. | 414 young adults (78.0% female; 19.6% male; 2.4% transgender), 18–34 years, $M_{age}$ =23.26, $SD$ =3.75, 93.5% White, 1.9% Asian, 4.3% other, 0.2% not reported. | Cross-sectional | UK | Age, depression, anxiety, preferential information processing (need for affect approach and avoidance, need for cognition). | SBQ-R [39] One item assesses lifetime ideation or attempt "Have you ever thought about or attempted to kill yourself". | Age, depression, depression × need for affect approach, anxiety × need for cognition, need for affect avoidance × need for cognition. | The SA group reported significantly greater levels of depression compared to the SI group (Cohen's $d$ =0.40). The SA group reported significantly greater age compared to the SI group (OR = 1.84, 95% CI 1.23–2.66). Three two-way preferences in information processing distinguished the SA group from the SI group: depression × need for affect approach (OR = 0.61, 95% CI 0.39–0.96), anxiety × need for cognition (OR = 0.56, 95% CI 0.35–0.89), and need for affect avoidance × need for cognition (OR = 1.61, 95% CI 1.12–2.31). |

*(Continued)*

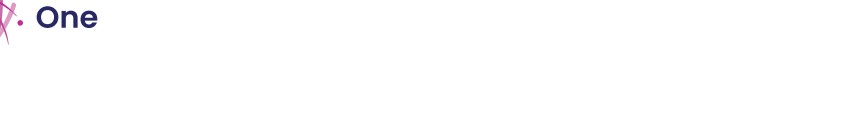

**Table 2.** (Continued)

| Author and date | Sample type and demographics | Study design | Country | Exposure: risk or protective factors measured | Suicide measure | Significant risk or protective factors for suicide attempts | Study findings |
|---|---|---|---|---|---|---|---|
| Cruz et al [58]. | 42 adolescents referred to outpatient departments (86% female), 13–21 years, $M_{age}$ = 16 years, $SD$ =1.86. Ethnicity not reported. | Cross-sectional | Portugal | Age, number of school failures, paternal and maternal parenting styles (parental rejection, parental control, and emotional support), attachment to mother and father (quality of emotional bond, separation anxiety, and dependence and inhibition of exploration and individuality), family functioning (cohesion and adaptability), satisfaction with familial relationships, and psychological symptoms (internalization- depression, internalization- anxiety, externalization- destructiveness, and externalization- exhibitionism). | Identified by their psychiatrist/ psychologists as having NSSI, SI, and SA. No measure specified. | Greater father rejection, lower mother control | The SA group reported significantly greater father's rejection (β=3.14**) and lower mother's control (β=−2.14*) than the SI group. Age did not significantly differ between the SA and SI group. |

*(Continued)*

| Author and date | Sample type and demographics | Study design | Country | Exposure: risk or protective factors measured | Suicide measure | Significant risk or protective factors for suicide attempts | Study findings |
|---|---|---|---|---|---|---|---|
| Dadras & Wang [59] | 5692 adolescents in Grades 7–12 (53.19% female), 11–18 years, $M_{age}$ = 14.60 years, $SD$ = 0.14. SI: $N$ = 765. SA: $N$ = 568. SA among those who had SI: $N$ = 360. | Cross-sectional | Lebanon | Age and sex. Physical and psychological harms: felt lonely, felt worried, was physical attacked, had a physical fight, was bullied (kicked or shoved; made fun of because of race; made fun of because of religion; made fun of about sex; left out of activities; made fun of body; bullied some other way), missed school, parental support, food insecurity. Substance-use related variables: ever/current mar-ijuana/ amphet-amine use/ age at first drug/ cigarette/ alcohol use, current use of cigarette, other tobacco products/ alcohol drinking, quantity of alcohol, got drunk/ troubled drunk. | SI: "During the past 12 months have you ever seriously considered attempting suicide?" SA: "During the past 12 months, how many times did you actually attempt suicide?" | Being female, always felt lonely, felt worried most of the time or always, physically attacked ≥ 2 times, engaged in ≥ 2 physical fight (all in last 12 months), bullied (kicked, pushed, or shoved; made fun of about sex), food insecurity sometimes or most of the time (past month), ≥6 days cigarette use, ≥ 6 days of other tobacco use per month, ≥14 age of cigarette initiation, alcohol consumption and quantity. | Females with SI were significantly more likely to attempt suicide than males (aOR = 1.35; 95% CI 1.02–1.78). Among those with SI, those who always felt lonely (aOR = 3.84; 95% CI 2.32–6.37) or always felt worried (aOR = 3.76; 95% CI 2.32–6.09) were significantly more likely to attempt suicide than those who never or rarely had these feelings. Among those with SI, those who had been physically attacked (aOR = 1.70, 95% CI 1.13–2.55) or engaged in a physical fight more than twice in the past 12 months (aOR = 2.20, 95% CI 1.43–3.38) were significantly more likely to attempt suicide compared to those who had not. Among those with SI, those who were bullied, specifically kicked, pushed, or shoved (aOR = 3.96, 95% 1.52–10.31) or made fun of about sex (aOR = 5.39, 95% CI 1.74–16.59) were significantly more likely to attempt suicide. Among those with SI, those who sometimes (aOR = 1.60, 95% CI 1.01–2.52) or most of the time (aOR = 2.71, 95% CI 1.24–5.91) experienced food insecurity. For substance use, among those with SI, ≥6 days cigarette use (aOR = 3.71, 95% CI 2.15–6.39), ≥6 days of other tobacco use per month (aOR = 3.06, 95% CI 1.91–4.90), ≥14 age of cigarette initiation (aOR = 2.53, 95% CI 1.02–6.26), alcohol consumption (1–5 days/ month aOR = 1.90, 95% CI 1.15–3.14) and quantity (≥ 6 days/ month aOR = 2.36, 95% CI 1.27–4.39) were significantly associated with suicide attempts. |

**Table 2.** (Continued)

| Author and date | Sample type and demographics | Study design | Country | Exposure: risk or protective factors measured | Suicide measure | Significant risk or protective factors for suicide attempts | Study findings |
|---|---|---|---|---|---|---|---|
| Dean-Boucher et al [60]. | 10,148 adolescents, 13–18 years. Sex/ gender, $M_{age}$ and SD not reported. | Cross-sectional | US | Chronic medical conditions: diabetes, epilepsy, arthritis, asthma, allergies, headache, back and neck pain, other pain, cardiovascular diseases, dermatological diseases, gastrointestinal diseases. | Computer-assisted CIDI 3.0 interview. SI: "Have you ever experienced seriously thinking about killing yourself?". SA: measure to assess SA is not reported. | Cardiovascular disease | Cardiovascular disease was the only chronic medical condition which distinguished the SA group from the SI group (aOR = 4.8, 95% CI 2.5–9.0), even when controlling for mental health disorders (aOR = 4.1, 95% CI 2.5–9.0). |
| Delfabbro et al [61]. | 2552 secondary school students (58.20% female; 1.02% undisclosed), $M_{age}$ =15.2 years, $SD$=0.5. | Cross-sectional | South Australia | Gender, smoking, being in a romantic relationship. Social variables: Being bullied by peers, being bullied by teachers, number of close friends, financial security, family functioning, number of peers they do not like, and number of class peers that disliked them. Psychological variables: Negative mood, self-esteem, General Health Questionnaire-12, anomie, and life satisfaction. Health-related variables: Weight status, used marijuana. | Scale was devised for the current study. Participants were asked if they had ever had thoughts of killing themselves (some SI), ever had these thoughts persistently (persistent SI), ever made plans to kill themselves (suicidal plan), and ever attempted suicide (SA). | Female gender, being a smoker, being in a romantic relationship, being bullied by peers, greater score on the General Health Questionnaire-12, negative mood, anomie, lower self-esteem and life satisfaction. | In comparison to the group experiencing some SI, the SA group were more likely to report being involved in a romantic relationship (OR = 3.15, 95% CI 1.81–5.47). In comparison to the group experiencing some SI, the SA group were less likely to report being male (OR = 0.45, 95% CI 0.24–0.84) and being a non-smoker (OR = 0.38, 95% CI 0.20–0.72). In comparison to the group experiencing some and persistent SI, the SA group were more likely to report being bullied by peers and greater scores on the General Health Questionnaire-12 (statistics not reported). The SA group were also more likely to report greater negative mood and anomie, and lower self-esteem and life satisfaction, compared with the group experiencing some SI (statistics not reported). |

*(Continued)*

| Author and date | Sample type and demographics | Study design | Country | Exposure: risk or protective factors measured | Suicide measure | Significant risk or protective factors for suicide attempts | Study findings |
|---|---|---|---|---|---|---|---|
| De Luca et al [62]. | 16,600 adolescents (retained through to Wave 4). 57.83% non-Hispanic white (50.87% female; $M_{age}$ =16.2 years), 22.78% black non-Hispanic (52.71% female; $M_{age}$ =16.66 years), and 19.40% Latino (50.12% female; $M_{age}$ =17.11 years),. | Longitudinal | US | Help-seeking | Measured using a categorical variable based on three severity levels of suicidal behaviours: (1) any SA in the past 12 months or if the attempt resulted in an injury, poison-ing, or overdose needing medical treatment (i.e., only SA); (2) SI, including if they reported seriously considering suicide in the past 12 months (i.e., only SI); and (3) neither a disclosure of SI/ SA). | Help-seeking | For black males who disclosed SI at Wave I, seeking help (OR= 0.25, $p$=0.009) at Wave I reduced 75% of the odds of SA at Wave III. For Latinas with SI, seeking help at Wave I had a 91% decrease in odds of attempts at Wave III (OR = 0.089, $p$ = 0.030). No other findings were significant. |
| Eisenlohr-Moul et al [63]. | 220 female adoles-cents, aged 12–16 years, $M_{age}$ = 14.69, $SD$=1.37, 64% White, 24% African American, 10% mixed/other, 1% Asian American, and 1% Latino/a. | Longitudinal | US | Trier Social Stress Test cortisol responses | SITBI [45]. SI: "Have you ever had thoughts of killing yourself?". SA: "Have you ever made an actual suicide attempt with at least some intent to die?" | Blunted cortisol reactivity | Individuals with lifetime SA experienced a blunted cortisol response across the four Trier Social Stress Test samples compared to those reporting a lifetime history of SI only (Interaction Estimate at Peak= −0.027, SE=0.013, $t$(689) = −2.07, p=0.038). |
| Fang [64] | 12,920 adolescent students. Age and sex/ gender not reported. | Longitudinal | US | School-level house-hold income. | Items from the Add Health study. SI: "During the past 12 months, did you ever seriously think about committing suicide?". SA: "During the past 12 months, how many times have you ever attempted suicide?" | Being a boy in a low-income school (compared to boys in middle- or high-income schools). | Among those with SI, boys from middle (OR = 0.41) or high-income schools (OR = 0.46) were much less likely to attempt suicide compared to boys from low-income schools. |

**Table 2.** (Continued)

| Author and date | Sample type and demographics | Study design | Country | Exposure: risk or protective factors measured | Suicide measure | Significant risk or protective factors for suicide attempts | Study findings |
|---|---|---|---|---|---|---|---|
| Fekadu et al [65] | Adults aged ≥18 years. 489 young adults. Data included is for the aged <25 group. Sex/gender, $M_{age}$ and $SD$, and ethnicity not reported. | Cross-sectional, population-based. | Ethiopia | Marital status, education, depression, alcohol use disorder. | The World Mental Health Survey Initiative version of the WHO CIDI was adapted to assess the 12-month prevalence of suicidal behaviour. Single item measures. SI: "Have you thought of taking your life in the past 12 months?"; Suicide plan: "Did you ever make a plan for taking your own life at any time in the past 12 months?". SA: "Have you attempted to take your own life in the past 12 months?". | Depression (scoring 10 or above on the PHQ). | For participants in the <25 group, the only factor that distinguished between SI and SA was scoring 10 and above on the PHQ (OR = 1.83, 95% CI 0.52, 6.48). |
| Florez et al [66] | 580 youth (63.8% female), aged 10–17 years, $M$ = 14.28, $SD$ = 2.00, 58.3% White, 22.1% Hispanics, 17.4% Black/African American. | Cross-sectional | US | Age, gender, race, number of inpatient hospitalizations, history of homelessness, overall life functioning variables (family, living situation, social, recreational, legal, medical, physical sleep, school behaviour, achievement, and attendance), anxiety, depression, oppositional disorder, conduct disorder, somatization, trauma adjustment, impulsivity, and anger control. | C-SSRS [67], a semi-structured interview that assesses recent (i.e., one month) and lifetime SI and SA. | Older age, female, number of inpatient hospitalizations, impaired legal functioning, greater impulsivity. | Being older (OR = 1.14, $p$ = 0.02) and female (OR = 1.87, $p$ = 0.007) significantly predicted lifetime suicide attempts. Legal functioning (OR = 1.52, $p$ = 0.039), impulsivity ($\chi^2$ (1, $N$ = 564) = 7.42, $p$ = 0.006, OR = 1.47), and number of previous inpatient hospitalizations ($\chi^2$ (1, $N$ = 564) = 30.61, $p$ < 0.01, OR = 2.28) significantly predicted lifetime suicide attempts. |

*(Continued)*

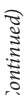

| Author and date | Sample type and demographics | Study design | Country | Exposure: risk or protective factors measured | Suicide measure | Significant risk or protective factors for suicide attempts | Study findings |
|---|---|---|---|---|---|---|---|
| Gatta et al [68]. | 190 adolescent inpatients (80.5% female), 10–19 years, $M_{age}$ = 14.5 years, $SD$ = 1.63. Ethnicity not reported. SI group: ($N$ = 97). SA group: ($N$ = 93). | Retrospective, Cross-sectional. | Italy | Age, gender, psychiatric familiarity, NSSI, impulsivity, problem behaviour (total, internalizing, externalizing behaviours), emotion dysregulation, and alexithymia. | Medical records and clinical interviews (not specified). | Gender, psychiatric familiarity, externalising problems (rule-breaking behaviour and conduct disorder), alexithymia. | Being female significantly distinguished the SA group from the SI group ($\chi 2$ = 6.79; $p$ = .009). Psychiatric familiarity ($\chi 2$ = 8.34; $p$ = .004), rule-breaking behaviour ($\chi 2$ = 9.04; $p$ = .011), conduct disorder ($\chi 2$ = 7.72; $p$ = .021), and alexithymia ($\chi 2$ = 8.16; $p$ = .017) significantly distinguished the SA group from the SI group. However, age (statistic not reported), NSSI ($\chi 2$ = 1.40; $p$ = .497), impulsivity ($\chi 2$ = 3.72; $p$ = .156), and emotion dysregulation ($\chi 2$ = 1.78; $p$ = .411) did not significantly distinguish the SA group from the SI group. |
| Georgia-des et al [69]. | 2396 youth (50.38% female) aged 14–17 years. SI group: (67.92% female; $M_{age}$ = 15.62, $SD$ = 0.17). SA group: 61.51% female; $M_{age}$ = 15.86, $SD$ = 0.15). Ethnicity was not reported. | Cross-sectional | Canada | Age, gender, sociodemographic variables (number of biological parents in home, household income below the low-income threshold, immigrant background, urban-rural residency), mental disorders (depression, any anxiety disorder, oppositional defiant or conduct disorder, ADHD), NSSI, cannabis or illicit substance use, tobacco use, heavy episodic drinking, peer victimization, exposure to child maltreatment. | Single-item questions: SI: "In the past 12 months, did you ever seriously consider taking your own life or killing yourself?" SA: "In the past 12 months, how many times did you actually try to take your own life?" | NSSI | Only NSSI significantly distinguished the SA group from the SI group (aOR = 3.89, 95% CI 1.45–10.43). Age (aOR = 1.10, 95% CI 0.71–1.70), female gender (aOR = 0.88, 95% CI 0.29–2.68), immigrant background (aOR =.039, 95% CI 0.14–1.06), cannabis or illicit substance use (aOR = 1.75, 95% CI 0.58–5.28), tobacco use (aOR = 1.78, 95% CI 0.54–5.92), and heavy episodic drinking (aOR = 4.64, 95% CI 0.92–23.39) did not significantly distinguish the SA group from the SI group. |

*(Continued)*

**Table 2.** (Continued)

| Author and date | Sample type and demographics | Study design | Country | Exposure: risk or protective factors measured | Suicide measure | Significant risk or protective factors for suicide attempts | Study findings |
|---|---|---|---|---|---|---|---|
| Glenn et al [70]. | 97 adolescents from long-term psychiatric inpatient units (50.5% female), $M_{age}$ = 15.42 years, $SD$ = 0.93. 9.4% Caucasian, 9.3% African American, 11.3% other. | Cross-sectional | US | Borderline personality disorder features: affective instability, identity problems, negative relationships and self-harm. Negative emotionality, Axis I internalizing diagnoses. | SBQ-R [39] Examined lifetime SA and frequency of SI in the past year. | Greater affective instability | Greater affective instability significantly distinguished those engaging in SA from those with thoughts of SI, beyond demographic and negative emotion-ality covariates (OR = 1.07, $p$ = .03). No other borderline personality facets significantly predicted being in the SA group among those with SI ($p$ > .05). |
| Glenn et al [71]. | Combined sample 276 adolescents with a history of suicidal thoughts of behaviours (71% female), 13–19 years $M_{age}$ = 15.53, $SD$ = 1.34, 79.3% European American, 7.6% Asian, 2.2% African American, and 9.4% multiracial. Wave 1: $N$ = 149, Wave 2: $N$ = 127. | Longitudinal | US | NSSI, Implicit Association Test | SITBI [45] measures the presence and frequency of self-injurious and suicidal thoughts, plans, and behaviors over the individual's lifetime, past year, past month, and past week. BSS [41] examined past week SI. | NSSI | Adolescents in the SA group reported significantly more lifetime NSSI than the SI only group ($\chi^2$(1, $N$ = 276) = 5.90, $p$ = .02. The SA group did not exhibit signifi-cantly stronger implicit identification with death compared to adolescents who SI only ($t$(273) = .30, $p$ = .76, $d$ = .04). |
| Heelis et al [72]. | 45 young men and women 16–35 years, $M$ = 24.96 years, $SD$ = 4.70, 11% female, 40% White British. SI group, SA group, Control group. | Retrospec-tive, Cross-sectional | UK | Thwarted belong-ingness, perceived burdensomeness, acquired capability and painful and provocative events. | C-SSRS [53]. Assessed the presence of past and current SI (past 6 months) and behav-ior (past 2 years). | No significant differ-ences found. | No significant differences were found for thwarted belongingness, perceived burdensomeness, acquired capability and painful and provocative events, across the SI, SA and control groups. |

*(Continued)*

| Author and date | Sample type and demographics | Study design | Country | Exposure: risk or protective factors measured | Suicide measure | Significant risk or protective factors for suicide attempts | Study findings |
|---|---|---|---|---|---|---|---|
| Hielscher et al [73]. | 2640 adolescents aged 12–17 years, $M_{age}$ =13.9, 72.1% female (Time 1). $N$=1975 (Time 2). SI sub-sample: $N$=216. | Longitudinal 1-year follow-up | Australia | Auditory hallucinatory experiences, psychological distress, three-way interaction of perceived burdensomeness, thwarted belongingness, and acquired capability. | Self-Harm Behaviour Questionnaire [74]. Single item measures. SI: "Have you ever thought about ending your life?" SA: "Did you ever try to end your life?". | Psychological distress, interaction of acquired capability and perceived burdensomeness, and interaction of auditory hallucinatory experiences and psychological distress. | The direct effect of baseline psychological distress on future suicide attempts (b=1.41, SE=0.69, $p$=0.04; OR = 4.09, 95% CI 1.07–15.68) was significant. The direct effect of the two-way (acquired capability x perceived burdensomeness) interaction at T1 (not T0) on incident suicide attempts (b=0.10, SE=0.04, $p$=0.02; OR = 1.09, 95% CI=1.01–1.17) was significant. However, the association between perceived burdensomeness and incident suicide attempts was only significant at the highest level of acquired capability for suicide. The interaction of auditory hallucinatory experiences and psychological distress significantly predicted suicide attempts one year later (OR = 9.58, 95% CI 1.61, 56.98). |
| Hong et al [75]. | 66 adolescents/ young adults diagnosed with MDD aged 14–25 years. SI group without a lifetime history of SA: $N$=25 (38.89% female), $M_{age}$ =18.38 years, $SD$=3.94. SA group: $N$=41 (46.4% female), $M_{age}$ =19.54 years, $SD$=2.86. | Cross-sectional | China | Sex, age, education level, depression, and 10 top-ranked features identified by SVM-RFE: Right lateral orbitofrontal thickness, left caudal anterior cingulate thickness, left fusiform thickness, left temporal pole volume, right rostral anterior cingulate volume, left lateral orbitofrontal thickness, left posterior cingulate thickness, right pars orbitalis thickness, right posterior cingulate thickness, and left medial orbitofrontal thickness. | Review of medical records and interview using the C-CASA [76]. Examined history of SA within the past 6 months in the SA group. Current SI and lifetime history of SA was measured in the SI group. | Depression, MRI data: Right lateral orbitofrontal thickness, left fusiform thickness, left temporal pole volume, left lateral orbitofrontal thickness, left posterior cingulate thickness, right pars orbitalis thickness, right posterior cingulate thickness, and left medial orbitofrontal thickness. | The SA group had significantly higher depression scores than the SI group ($p$=0.023). There were no significant differences between groups with respect to age, sex ratio and education level. Right lateral orbitofrontal thickness ($t$=−5.998, $p<0.001$), left fusiform thickness ($t$=−5.323, $p<0.001$), left temporal pole volume ($t$=−5.580, $p<0.001$), left lateral orbitofrontal thickness ($t$=−6.040, $p<0.001$), left posterior cingulate thickness ($t$=−3.397, $p<0.01$), right pars orbitalis thickness ($t$=−4.672, $p<0.001$), right posterior cingulate thickness ($t$=−3.416, $p<0.01$), and left medial orbitofrontal thickness ($t$=−3.508, $p<0.01$) were significantly greater in those with SI only, compared to the SA group. The left caudal anterior cingulate thickness and right rostral anterior cingulate volume did not significantly distinguish between SI and SA groups. |

*(Continued)*

| Author and date | Sample type and demographics | Study design | Country | Exposure: risk or protective factors measured | Suicide measure | Significant risk or protective factors for suicide attempts | Study findings |
|---|---|---|---|---|---|---|---|
| Ivanich et al [77]. | 46 American Indian youth (70% female), $M_{age}$ = 16.33, $SD$ = not reported. | Cross-sectional | US | Proportion of network that expressed suicide, uses alcohol and drugs, used alcohol and drugs with. Proportion of network participants learnt apache with/did apache activities with. Proportion of network participants' feel listens, treat you well, helped with emotional problems, received help for SI, exposed to suicide, hurt self and have problems with people. Proportion of network that are native, have Christian beliefs with, are cousins, in school, same age, same sex, elders, are in a gang, that are considered a caretaker, considered a role model and participants bullied. Average trust scores, average length of years known alters. | No measure reported. All cases who were reported to the surveillance system for a SA or SI were approached by community mental health specialists for participation in the study. | Greater proportion of network uses alcohol, greater proportion of network uses drugs, greater proportion of network they use alcohol with, greater proportion of network they use drugs with, more likely to reach out for help when struggling. | The suicidal ideation group reported a significantly lower proportion of their network that uses alcohol ($t$ = 2.18; $p < 0.05$), uses drugs ($t$ = 2.94; $p < 0.05$; that they use alcohol with ($t$ = 2.75; $p < 0.01$) and that they use drugs with ($t$ = 2.66; $p < 0.05$) compared to the suicide attempt group. The suicide attempt group reported that they reached out to a greater number of alters for help when they were struggling with suicide compared to the suicidal ideation group ($t$ = 2.65; $p < 0.05$). Other factors included that reached marginal significance were average length of time participants have known their alters ($p = 0.06$), proportion of alters that have hurt themselves ($p = 0.09$), and proportion of alters in a gang ($p = 0.068$). |
| Jenkins et al [78]. | Youth aged 14–24 years. NSSI and SI group with no history of SA: $N$ = 77 (76.62% female), $M_{age}$ = 17.87 years, $SD$ = 2.77, 81.58% White. NSSI and SA group: $N$ = 56 (69.64% female), $M_{age}$ = 18.13 years, $SD$ = 2.71, 69.64% White. | Cross-sectional | US | Age, sex, ethnicity, depression, anxiety, PTSD, substance abuse, eating disorder. | The Behavioural Health Screen [79]. | Substance use | Youth in the NSSI + SA group reported significantly greater substance use than the NSSI + SI group. |

| Author and date | Sample type and demographics | Study design | Country | Exposure: risk or protective factors measured | Suicide measure | Significant risk or protective factors for suicide attempts | Study findings |
|---|---|---|---|---|---|---|---|
| Khoubaeva et al [80]. | 197 adolescents diagnosed with bipolar disorder, 13–20 years. SI only group: $N$=29 (51.7% female), $M_{age}$ = 16.84 years, $SD$= 1.31, 89.7% Caucasian. SA (with or without NSSI) group: $N$=41 (80.5% female), $M_{age}$ = 16.64 years, $SD$= 1.43, 80.5% Caucasian. NSSI (with or without SI) group: $N$=75 (81.3% female), $M_{age}$ = 16.52 years, $SD$= 1.59, 84% Caucasian. No SI/SA/NSSI group: $N$=52 (46.2% female), $M_{age}$ = 16.93 years, $SD$= 1.51, 71.2% Caucasian. | Cross-sectional, retrospective | Canada | Age, sex, SES, race, living with both biological parents, bipolar subtypes (BD-I, BD-II, BD-NOS), mood symptom severity and functioning, lifetime comorbid diagnoses (ADHD, any anxiety, PTSD, OCD, eating disorder, SUD, ODD and conduct disorder), other characteristics (smoking, lifetime sexual abuse, lifetime physical abuse, lifetime psychosis, and legal history), family psychiatric history (manic/hypomania, major depressive episode, SI, SA, anxiety, and ADHD), and lifetime treatment history (second-generation antipsychotic, lithium, SSRI antidepressants, non-SSRI antidepressants, stimulants, psychiatric hospitalization), affective regulation, and life problems (impulsivity, emotion dysregulation, identify confusion and interpersonal problems). | SI: Systematically queried and recorded on a Safety Assessment form. SA: The Adolescent Longitudinal Follow-up Interview suicide attempt form (capturing method, medical threat, and intent of each instance of self-harm). | Female sex, eating disorders, major depressive episode, adolescent affect regulation, emotion dysregulation and identify confusion. | Those in the SA group were significantly more likely to be female to report eating disorders, major depressive episode, adolescent affect regulation, impulsivity and emotion dysregulation, compared to the SI group. |

(Continued)

| Author and date | Sample type and demographics | Study design | Country | Exposure: risk or protective factors measured | Suicide measure | Significant risk or protective factors for suicide attempts | Study findings |
|---|---|---|---|---|---|---|---|
| Kim et al [81]. | 73,238 adolescents aged 12–19 years (47.6% female). | Cross-sectional | Korea | Age, sex, low SES, living with a single parent or others, heavy alcohol use, heavy smoking, drug use, academic achievement, perceived stress level, unhealthy coping strategy, high perceived sadness/ hopelessness. | Single item measures from the 2010 Korea Youth Risk Behavior Web-Based Survey. SI: "During the past 12 months, did you ever seriously consider attempting suicide?" SA: "During the past 12 months, did you ever actually attempt suicide?" | Heavy alcohol use, drug use, high perceived sadness/ hopelessness, low academic achievement, poor perceived health status, high perceived stress level, unhealthy coping strategy, younger age (12–15 years; females only), and living with single parent or others (females only). | In the total sample, being female (OR = 1.19, 95% CI 1.06–1.33), younger age (12–15 years; OR = 1.12, 95% CI 1.01–1.25), heavy alcohol use (OR = 1.61–1.30, 1.99), drug use (OR = 4.40, 95% CI 2.95–6.55), low academic achievement (OR = 1.32, 95% CI 1.19–1.46), poor perceived health status (OR = 1.29, 1.14–1.45), high perceived stress level (OR = 1.12, 95% CI 1.01–1.25), unhealthy coping strategy (OR = 1.57, 95% CI 1.30–1.90), and high perceived sadness/ hopelessness (OR = 1.48, 95% CI 1.30–1.69) distinguished the SA group from the SI group. In males only, heavy alcohol use (OR = 2.06, 95% CI 1.56–2.72), drug use (OR = 3.45, 95% CI 2.11–5.65), and perceived sadness/ hopelessness (OR = 1.51, 95% CI 1.24–1.83) distinguished the SA group from the SI group. In females only, younger age (12–15 years, OR = 1.16, 95% CI 1.01–1.33), living with a single parent or others (OR = 1.23, 95% CI 1.06–1.43), heavy alcohol use (OR = 1.54, 95% CI 1.34–1.78), drug use (OR = 6.95, 95% CI 3.47–13.91), low academic achievement (OR = 1.41, 95% CI 1.24–1.60), poor perceived health status (OR = 1.30, 95% CI 1.13–1.49), high perceived stress level (OR = 1.37, 95% CI 1.17–1.60), unhealthy coping strategy (OR 2.14, 95% CI 1.61–2.86) and high perceived sadness/ hopelessness (OR = 1.46, 95% CI 1.23, 1.73) distinguished the SA group from the SI group. No other variables significantly distinguished the SA group from the SI group. |

*(Continued)*

Table 2. (Continued)

| Author and date | Sample type and demographics | Study design | Country | Exposure: risk or protective factors measured | Suicide measure | Significant risk or protective factors for suicide attempts | Study findings |
|---|---|---|---|---|---|---|---|
| Kwon et al [82]. | 2258 middle and high school students (44.9% female), $M_{age}$ = 15.63 years, SD not reported. | Cross-sectional | Korea | Depressive symptoms, impulsivity, poor family relationship, social support (from parents, teachers, and peers), bullying behaviour, peer-victimization, alcohol/drug use, internet related delinquency, life stressors, self-esteem, attitudes towards suicide, NSSI, and sexual abuse. | Single item measures. SI: "Have you seriously thought of committing suicide in the past 12 months?" SA: "Have you attempted to commit suicide in the past 12 months?" | NSSI. | The SA group reported greater NSSI compared to the SI group. |
| Kwon et al [31]. | 7498 adolescents experiencing SI, 12–18 years. No SA group: N = 5960 (62.0% female), adolescents aged 12–15 years (N = 3188), adolescents aged 16–18 years (N = 2301). SA group: N = 1538 (66.1% female), adolescents aged 12–15 years (N = 890), adolescents aged 16–18 years (N = 466). | Cross-sectional | Korea | General factors: gender, age, academic performance, economic level, type of residence, number of breakfasts per week, hours of smartphone use on weekdays and weekends. Physical factors: use of alcohol, smoking, habitual or deliberate drug use, sexual experience, experience of being treated for violence in the last 12 months, degree of fatigue recovery from sleep for the last 7 days, body mass index (BMI), efforts for weight control, and number of days per week of physical activity for more than 60 min a day for the last 7 days. Psychological factors: experiencing stress and the presence of depression within two weeks. | Not reported. | General factors: Being female, lower age, lower academic performance and lower economic level. Physical factors: Smoking, alcohol drinking, drug experience, sexual experience, violent treatment experience, reduction in weight control effort, physical activity. Psychological factors: depression. | For general factors, there were significantly lower rates of suicide attempts among adolescents with SI who were male (OR = 0.74, 95% CI 0.60–0.90) and significantly higher rates among adolescents with SI who were of a lower age (12–15 years; OR = 1.64, 95% CI 1.36–1.98) had lower academic performance (OR = 1.24, 95% CI 1.01–1.52) and were of a lower economic level (OR = 1.24, 95% CI 1.01–1.53). For physical factors, there were significantly higher rates of suicide attempts among adolescents with SI who smoked (OR = 1.38, 95% CI 1.12–1.71), drank alcohol (OR = 1.305, 95% CI 1.08–1.58), used drugs (OR = 3.84, 95% CI 2.517–5.863), had sexual experience (OR = 1.60, 95% CI 1.21–2.12), received treatment for violence (OR = 3.34, 95% CI 2.39–4.66), reduction in weight control effort (OR = 1.22, 95% CI 1.02–1.46) or engaged in more physical activity (OR = 1.28, 95% CI 1.00–1.63). For psychological factors, there were significantly higher rates of suicide attempts among adolescents with SI who experienced depression (OR = 2.0 95% CI 1.59–2.52). |

*(Continued)*

| Author and date | Sample type and demographics | Study design | Country | Exposure: risk or protective factors measured | Suicide measure | Significant risk or protective factors for suicide attempts | Study findings |
|---|---|---|---|---|---|---|---|
| Kyron et al [83]. | 2655 young people (48.6% female) aged 12–17 years, $M_{age}$ and SD not reported. SI group: N=146 (69.4% female). SA group: N=74 (68.7% female). | Cross-sectional | Australia | Number of mental health disorders, MDD in the last year, feeling hopeless, conduct problems, peer problems, family functioning, low school connectedness, NSSI, ever had sexual intercourse, binge drinking in the past 30 days, weekly cigarette smoking, and illicit drugs in the past 30 days. | SI: Adolescents were asked whether they had seriously thought about taking their own life in the past 12 months. SA: Adolescents were asked whether they had ever attempted suicide, or within the prior 12 months. | Number of mental health disorders, poor family functioning, conduct problems, weekly cigarette smoking, ever experienced sexual intercourse. | Among those with SI (without substance use factors), number of mental health disorders (β=1.8, 95% CI 1.2–2.6), conduct problems (OR = 2.7, 95% CI 1.2–6.2), poor family functioning (OR = 6.1, 95% CI 1.4–27.7), and ever having sexual intercourse (OR = 2.5, 1.00–6.0) were associated with higher rates of suicide attempts. In the model which excluded 12-year-old participants (without substance factors), number of mental health disorders (OR = 1.7, 95% CI 1.2–2.5), conduct problems (OR = 2.4, 95% CI 1.0–5.5), and poor family functioning (7.3, 95% CI 1.7–32.3) remained significant. In the model which included substance use factors, number of mental health disorders (OR = 1.7, 95% CI 1.1–2.6), poor family functioning (OR = 10.1, 95% CI 2.3–44.7), and weekly cigarette smoking (OR = 4.5, 95% CI 1.7–12.0) predicted suicide attempts among those with SI. No other variables examined significantly predicted suicide attempts among those with SI. |
| Lear et al [84]. | 5996 10th and 12th grade high school students (49.60% female), $M_{age}$=16.51, SD=1.12, 76.70% White and of non-Hispanic origin (83%). | Cross-sectional | US | Age, sex, race, changed school in the past year, felt unsafe at school, being bullied, alcohol use, cigarette smoking, early initiation of drug use, total lifetime drugs, sensation seeking, psychological distress, family involvement, community disorganization. | Single item measures. SI: "During the past 12 months, did you ever seriously consider attempting suicide?". SA: "During the past 12 months, how many times did you actually attempt suicide?" | Middle school students only: Being female, non-White race, regularly feeling unsafe at school, experiencing bullying over the past 12 months, higher lifetime drug use, increased community disorganization, increasing community disorganization and psychological distress. High school students only: Younger age, earlier use of drugs, having changed schools, experienced bullying in the past year. | In middle school students reporting SI, being female (OR = 1.45, p=.010), of non-White race (OR = 1.57, p<.001), feeling unsafe at school (OR = 1.49, p=.005), experiencing bullying in the past 12 months (OR = 1.39, p=.021), greater lifetime drug use (OR = 1.29, p=.026), increased community disorganization (OR = 1.20, p=.004) and psychological distress (OR = 1.50, p<.001) significantly predicted suicide attempts in the past year. In high-school students, younger age (OR = 0.88, p=.040), earlier use of drugs (OR = 1.36, p=.002), having changed schools (OR =1.69, p=.005) being bullied in the past year (OR = 1.97, p<.001), and psychological distress (OR = 1.33, p<.001) predicted suicide attempts in the past year. |

*(Continued)*

**Table 2.** (Continued)

| Author and date | Sample type and demographics | Study design | Country | Exposure: risk or protective factors measured | Suicide measure | Significant risk or protective factors for suicide attempts | Study findings |
|---|---|---|---|---|---|---|---|
| Li et al [85]. | 1755 adolescents (65.1% female), 48.4% White, 12.6% Black, 7.1% Hispanic, 31.9% Other. No physical fighting/sexual violence: $N$=827. Sexual violence without physical fighting: $N$=360. Physical fighting without sexual violence: $N$=318. Physical fighting and sexual violence: $N$=250. | Cross-sectional | US | Physical fighting, sexual violence. | Single item measures from the 2019 Youth Risk Behaviour Survey. SI: "During the past 12 months, did you ever seriously consider attempting suicide?" SA: "During the past 12 months, how many times did you actually attempt suicide?". | Interaction of physical fighting and sexual violence | The concurrence of physical fighting and sexual violence substantially increased the relative risk of attempted suicide (aRR = 1.99, 95% CI 1.72, 2.29) |
| Liu et al [86]. | 569 adolescents (86.3% female), $M_{age}$ =21.94 years, $SD$=3.31. SI group: $N$=277 (85.6% female), $M_{age}$ =22.07 years, $SD$=3.59. SA group: $N$=292 (87.0% female), $M_{age}$ =21.82 years, $SD$=3.02. | Cross-sectional | China | Sex, educational level, marital status, living status, suicide-related social media use behaviours, level of SI. | SI: 4-item measure of the ASIQ [87]. SA: Single item-measure. "Have you ever tried to kill yourself?". | Level of SI, suicide-related social media use behaviours, which include attending to suicide information, commenting on or reposting information, or talking about suicide. | The SA group reported a significantly higher level of SI ($t_{563.64}$ = 5.04; $p$ < .001) than the SI group. The SA group reported more *attending to* ($t_{567}$ = 1.94; $p$ = .05), *commenting-reposting* ($t_{567}$ = 2.12; $p$ = .03), and *talking about* ($t_{542.22}$ = 5.12; $p$ < .001) suicide-related social media use behaviours than the SI group. |
| Liu et al. [88] | 2095 substance-using adolescents with clinically significant depressive symptoms (75.4% female), aged 12–17 years ($M_{age}$ = 15.70, $SD$ = 0.04). 70.7% non-Hispanic white, 19.0% Hispanic, 4.6% non-Hispanic black, 3.0% multiracial, 2.1% Asian, Hawaiian, or Pacific Islander, and 0.6% Native American. 81.6% of participants had a history of SI, and 37.0% had a history of SA. Among adolescents with SI, 45.4% made a SA. | Cross-sectional | US | Age, gender, race/ethnicity, family income, depressive symptoms, cocaine, heroin and stimulant abuse and dependence symptoms. | Measured using items adapted from the depression section of The National Comorbidity Survey-Replication [89] and the National Comorbidity Survey Replication Adolescent Supplement [90,91]. | Depressive symptoms, Non-Hispanic Black race, family income of <$20,000 or $50,000-$70,000, injection substance use. | Among those with SI, being of Non-Hispanic Black race (OR = 0.39, 95% CI 0.19–0.82) were significantly less likely to attempt suicide. Among those with SI, having a family income of <$20,000 (OR = 1.95, 95% CI 1.34–2.85) or $50,000-$70,000 (OR = 1.50, 95% CI 1.03–2.17) were significantly associated with suicide attempts. Among those with SI, depressive symptoms was significantly associated with suicide attempts (OR = 1.31, 95% CI 1.16–1.47). Among those with SI, injection substance use was significantly associated with suicide attempts (OR = 2.89, 95% CI 1.68–4.98). |

*(Continued)*

| Author and date | Sample type and demographics | Study design | Country | Exposure: risk or protective factors measured | Suicide measure | Significant risk or protective factors for suicide attempts | Study findings |
|---|---|---|---|---|---|---|---|
| Liu et al [92]. | SI group: $N$=16,969, $M_{age}$ =19.6 years, $SD$=1.73, 65.3% female, 89.4% Han ethnicity, 10.6% Han minority. SA group: $N$=976, $M_{age}$ =19.7 years, $SD$=1.86, 65.6% female, 87.9% Han ethnicity, 12.1% ethnic minority. | Cross-sectional | China | Childhood trauma (emotional abuse, emotional neglect, physical abuse, physical neglect, and sexual abuse). | Chinese version of the SBQ-R [39]. | Emotional abuse and emotional neglect | Emotional abuse (OR = 1.11, 95% CI 1.11–1.12) and emotional neglect (OR = 1.03, 95% CI 1.02–1.05) significantly distinguished the SA group from the SI group. However, physical abuse (OR = 1.01, 95% CI 1.00–1.02), physical neglect (OR = 1.02, 95% CI 1.01–1.03) and sexual abuse (OR = 1.00, 95% CI 0.99–1.01) did not distinguish between SA and SI groups. |
| Macryni-kola et al [93]. | 1712 (81% female) college students, aged 17–37 years, $M_{age}$ =22.76 years, $SD$=4.53. Five groups: SI group, SA group, SA and NSSI group, NSSI group, and no self-injurious thoughts or behaviours group. | Cross-sectional | US | Stressful life events, social connectedness. | SI and SA questions adapted from the Columbia Suicide Screen [94]. SI: "Have you ever thought about killing yourself?" and "If yes, did you think about it during the last 12 months?". SA: "Have you ever tried to kill yourself?" and "If yes, did you make an attempt in the last 12 months?". | No factors were significant. | There were no significant differences in the number of stressful life events and social connectedness among SI and SA groups. |
| Marengo et al [36]. | 183 psychiatric patients (49.1% female) aged 18–30 years, $M_{age}$ = 23.3, $SD$=3.7 years, 74.8% White. SA group: $N$=67 (56.7% female), $M_{age}$ = 23.3 years, $SD$=3.7, 64.1% White. SI group: $N$=74 (46.0% female), $M_{age}$ = 24.3 years, $SD$=4.1, 80.3% White. Psychiatric controls: $N$=42 (36.6% female), $M_{age}$ =23.9 years, $SD$=3.4, 6.1% White. Healthy controls: $N$=40 (55.0% female), $M_{age}$ =24.9 years, $SD$=3.6, 87.5% White. | Longitudinal (Baseline, 3 months, 6 months, and 12 months). | US | Marijuana use, opioid use. | C-SSRS [53]. Severity of suicide ideation was measured with the SIQ [87]. Longitudinal Interval Follow-Up Evaluation and medical records were used to track suicidal behaviour prospectively. | Marijuana use in the past month, opioid use in the past month. | The SA group (β =1.31, 95% CI 0.4–2.3, $p$=0.01) and psychiatric controls were more likely than all other groups to have used marijuana in the past month. Controlling for sex, age, race, lifetime history of SA, including baseline attempt for the SA group, and clinical characteristics that were significantly associated with SA at follow-up, opioid use in the past month at baseline significantly predicted SA at 6 months (OR = 1.87, 95% CI 1.1–3.3, $p$=0.03). |

| Author and date | Sample type and demographics | Study design | Country | Exposure: risk or protective factors measured | Suicide measure | Significant risk or protective factors for suicide attempts | Study findings |
|---|---|---|---|---|---|---|---|
| Mars et al [11]. | 456 adolescents who reported SI at baseline (aged 16 years; Time 1, Sex/ gender, $M_{age}$ and SD not reported. Of 456 adolescents, 310 completed measures at age 21 years (Time 2). (74.19% female), $M_{age}$ =20.92 years, SD=0.52, 95.72% White British, 4.28% Other. Subsample reporting a SA at follow-up: N=38 (73.68% female), $M_{age}$ =20.70 years, SD=0.54. Ethnicity not reported. | Longitudinal | UK | Sex, intelligence quotient, executive function, impulsivity, sensation seeking, personality traits, exposure to self-harm in others, life events, early adversity, body dissatisfaction, sleep problems, psychiatric disorders, hopelessness, depressive symptoms, substance use (cannabis and other illicit drug use), suicidal plans, and NSSI. | Single item measures. SI: "Have you ever thought of killing yourself, even if you would not really do it?". SA (assessing self-harm and suicidal intent): "Have you ever hurt yourself on purpose in any way (e.g., by taking an overdose of pills or by cutting yourself)?" and "I wanted to die" and "On any of the occasions when you have hurt yourself on purpose, have you ever seriously wanted to kill yourself?". | NSSI, cannabis use, other illicit drug use, exposure to self-harm in others, higher levels of the personality type intellect/ openness. | 38 (12%) of 310 participants with SI reported that they attempted suicide for the first-time by follow-up (aged 21 years). The risk factors associated with SA among adolescents who reported SI at baseline included cannabis use (aOR = 2.61, 95% CI 1.11–6.14; p=0.029), other illicit drug use (aOR = 2.47, 95% CI 1.02–5.96; p=0.045), NSSI (aOR = 2.78, 95% CI 1.35–5.74; p=0.0059), and higher levels of the personality type intellect/openness (aOR = 1.62, 95% CI 1.06–2.46; p=0.025). There was also weak evidence of an association with exposure to self-harm in others (family member self-harm aOR 2.03, 95% CI 0.93–4.44, p=0.076; friend self-harm aOR = 1.85, 95% CI 0.93–3.69, p=0.081). |
| Mars et al [95]. | 4772 adolescents, (59.01% female), $M_{age}$ =16.68 years, SD=2.86, 96.14% White British, 3.86% Other. SI group: N=456; 73.25% female, $M_{age}$ =16.68 years, SD=2.77, 94.95% White British, 5.05% Other. SA group: N=325 (81.23% female), $M_{age}$ =16.70 years, SD =3.02, 95.50% White British, 4.50% Other. | Cross-sectional | UK | Sex, intelligence quotient (IQ), executive function, impulsivity, sensation seeking, personality traits, parental SA, parental cruelty, sexual abuse, being bullied, body dissatisfaction, psychiatric disorders, hopelessness, depressive symptoms, substance use (cannabis and other illicit drug use), heavy alcohol use, exposure to self-harm in friends and family, and life events. | Single item measures. SI: "Have you ever thought of killing yourself, even if you would not really do it?". SA (assessing self-harm and suicidal intent): "Have you ever hurt yourself on purpose in any way (e.g., by taking an overdose of pills or by cutting yourself)?", "I wanted to die", and "On any of the occasions when you have hurt yourself on purpose, have you ever seriously wanted to kill yourself?" | Female gender, exposure to self-harm in others (family member, friend, both family and friend), psychiatric disorder (depression, anxiety disorder, or behavioural disorder), lower intelligence quotient, impulsivity, intensity seeking, lower conscientiousness, life events, body dissatisfaction, hopelessness, smoking and illicit drug use (excluding cannabis). | Those in the SA group were significantly more likely to be female (aOR = 1.55, 95% CI 1.10–2.20), compared to the SI group. Factors that distinguished between adolescents with a history of SI and a history of SA were having a depressive disorder (aOR = 3.63, 95% CI 1.67–7.89), behavioural disorder, (aOR = 2.90, 95% CI 1.54–5.44), or anxiety disorder (aOR = 2.20, 95% CI 1.12–4.30), exposure to self-harm in others, including either family/friends (aOR = 3.21, 95% CI 2.14–4.82) or both friend and family self-harm (aOR = 5.26, 95% CI 3.17–8.74) and smoking (aOR = 2.54, 95% CI 1.61–4.02). The SA group were significantly more likely to report lower IQ (aOR = 0.80 95% CI 0.67–0.96), greater impulsivity (aOR = 1.19, 95% CI 1.01–1.42), greater intensity seeking (aOR = 1.17 95% CI 1.00–1.37), lower conscientiousness (aOR = 0.84, 95% CI 0.71–0.99), greater number of life events (aOR = 1.18 95% CI 1.04–1.33), body dissatisfaction (aOR = 1.70 95% CI 1.24–2.34), hopelessness (aOR = 1.47 95% CI 1.10–1.98), and illicit drug use (aOR = 1.80 95% CI 1.18–2.75) than the SI group |

*(Continued)*

| Author and date | Sample type and demographics | Study design | Country | Exposure: risk or protective factors measured | Suicide measure | Significant risk or protective factors for suicide attempts | Study findings |
|---|---|---|---|---|---|---|---|
| Masi et al [96]. | 41 adolescent inpatients, aged 11–18 years. SI group: $N=22$ (22.7% female), $M_{age}$ =14.86 years, $SD$ =1.86. SA group: $N=19$ (52.6% female), $M_{age}$ =15.05 years, $SD$ =1.75. | Cross-sectional | Italy | Age, gender, presence or absence of familial psychiatric disorders, familial attempted or completed suicides, familial depression, history of traumatic experiences, parental separation/divorce, bullying, family mourning, depression, negative mood, interpersonal problems, sense of ineffectiveness, anhedonia, low self-esteem, clinical disorders (ADHD, ODD, conduct disorder, OCD, psychotic symptoms, personality disorders, SUD, ASD, sleep disorders, eating disorders and learning disabilities), prevalent attitudes for life and death, impulsivity, hopelessness, resilience, and NSSI. | C-SSRS [53] assessed current SI and suicidal behaviour and severity of SI and suicidal behaviour. | Greater clinical severity, greater functional impairment, severe suicidal ideation with plan and intention, and greater duration of suicidal ideation. | The SA group reported significantly greater clinical severity (F =7.14 (1), $p$ =.011) and functional impairment (F =18.1 (1), $p$ <.001), compared to the SI group. Regarding the five types of SI based on the C-SSRS, the SA and SI groups significantly differed with the SA group reporting more severe ideation with plan and intention ($\chi^2$ =13.07 (5), $p$ =.023). The duration of SI was significantly greater in the SA group, compared to the SI group ($\chi^2$ =12.44 (5), $p$ =.029). No other variables significantly distinguished between the SA and SI groups. |

*(Continued)*

**Table 2.** (Continued)

| Author and date | Sample type and demographics | Study design | Country | Exposure: risk or protective factors measured | Suicide measure | Significant risk or protective factors for suicide attempts | Study findings |
|---|---|---|---|---|---|---|---|
| May et al [97]. | 7,433 students in grades 9–12 (63.9% female), aged 15–17 years. M and SD=not reported. 44.3% White. | Cross-sectional | US | Safety (e.g., driving behaviours, weapon carrying (in/out of school, and gun carrying), interpersonal violence (e.g., physical fight, bullying, dating violence), substance use (e.g., alcohol use, marijuana use, heroin use), sexual health (e.g., number of sexual partners) physical and mental health (e.g., asthma, sleep, physical activity), and use of electronics (hours of TV/ computer games), and study year. | Single item measures. SI: "During the past 12 months did you ever seriously consider attempting suicide?" SA: "During the past 12 months how many times did you actually attempt suicide?" | Heroin use, past year physical fights, youth who had experienced rape. | The classification tree was defined by: a lifetime history of rape, being in one or more physical fights in the past year, and ever using heroin. Four subgroups identified by terminal nodes include: (1)Those with SI who had never been forced to have sex, had been in a physical fight in the past year, and had ever used heroin had a very high probability of a SA (78%). (2)Those with SI who had ever been forced to have sex had a high probability of a SA (59%). (3) Those with SI who had never been forced to have sex, had been in a physical fight in the past year, and had never used heroin had a moderate probability of an SA (44%). (4) Those with SI who had never been forced to have sex and had not been in a physical fight in the past year had a low probability of a SA (29%). |
| McCallum et al [98]. | 1428 adolescent students (58% female, 1% other gender), aged 11–17 years, $M_{age}$ =13.3 years, SD=1.2. SI subgroup (N=192). | Cross-sectional | Australia | Age, gender, hopelessness, anxiety sensitivity, sensation-seeking, impulsivity, distress, gender*hopelessness, gender*anxiety sensitivity, gender*-sensation seeking, gender*impulsivity. | Single-item measures from the Youth Risk Behavior Survey. SI: "During the past 12 months, did you ever seriously consider killing yourself?". SA: "During the past 12 months, how many times did you actually try to kill yourself?". | Anxiety sensitivity | Anxiety sensitivity decreased the likelihood of suicide attempts (OR = 0.67, $p<0.05$) among the subgroup of adolescents experiencing suicide ideation. No other effects were significant. |

(Continued)

| Author and date | Sample type and demographics | Study design | Country | Exposure: risk or protective factors measured | Suicide measure | Significant risk or protective factors for suicide attempts | Study findings |
|---|---|---|---|---|---|---|---|
| McKay et al [32]. | 108 adolescents and young adults aged 14–21 years. Sex/ gender, $M_{age}$, $SD$, and ethnicity not reported. SI group: $N=70$ SA group $N=38$. | Cross-sectional | US | Sexual and gender minority status, skipping school due to feeling unsafe, feeling close to people at school, feeling that parents care, involvement in a LBGT+ group in the past 12 months, having accessed school or community-based mental health services in the past 12 months, and connectedness to other groups or teams in the past 12 months. | No specific measure reported. SI and SA were defined as seriously considered or attempted suicide within the past 12 months, respectively. | More likely to skip school due to feeling unsafe. | The SA group were significantly more likely to skip school due to feeling unsafe, compared to the SI group ($p<.01$). No other variables significantly distinguished between those in the SA or SI group. |
| Melhem et al [99]. | 115 psychiatric inpatients (43% female) aged 15–30 years, $M_{age}=23.0$ years, $SD=3.4$, 82% Caucasian. Patients were admitted following a SA, SI or were healthy controls. SI with no prior history of attempts group: $N=40$ (27.5% female), $M_{age}=23.6$ years, $SD=3.9$, 75% Caucasian. SA group: $N=38$ (44.7% female), $M_{age}=22.8$ years, $SD=3.8$, 89.5% Caucasian. Healthy controls: $N=37$ (54.1% female), $M_{age}=22.1$ years, $SD=2.2$, 83.8% Caucasian. | Cross-sectional | US | Severity of SI, childhood history of abuse, smoking, waist-hip circumference. Biological markers: Hair cortisol concentrations, expression of genes in the HPA axis and inflammatory pathways previously associated with suicidal behavior (GR or NR3C1, SKA2, FKBP5, IL–1β, TNF–α); plasma C-Reactive Protein (CRP); and cellular measures of glucocorticoid receptor (GR) sensitivity and stimulated production of IL–6. | Patients admitted for SA or SI. C-SSRS [53] assesses current and lifetime SI and SA, intensity of SI, and medical lethality of SA. | Severity of suicidal ideation, childhood history of abuse, smokers, lower waist-to-hip ratio. Lower hair cortisol concentrations, lower GR or NR3C1 (isoform) mRNA, higher CRP and higher TNF-α mRNA. | The SA group reported significantly higher severity of current SI ($t=6$, df=58.6, $p<0.001$) compared to the SI group. The SA group reported greater childhood history of abuse ($\chi^2=3.7$, df=1, $p=0.05$), were more likely to be smokers ($\chi^2=4.1$, df=1, $p=0.04$), and had lower waist-hip circumference ($t=-2.1$, df=76, $p=0.04$), compared to the SI group. However, these post-hoc differences did not reach statistical significance ($p<0.017$). The SA group exhibited significantly lower hair cortisol concentrations compared to the SI and control group, after controlling for significant covariates (β = –0.42, 95% CI –0.78–-0.05, $p=0.02$, ES= –0.38). The SA group had significantly lower GR–α mRNA compared to the SI and control group (β= –4.7, 95% CI –9.45–0.02, $p=0.05$, ES= –0.40). The SA showed significantly higher CRP (β=0.99, 95% CI 0.15–1.84, $p=0.02$, ES=0.64) and TNF–α, mRNA (β=16.8, 95% CI 1.9–31.7), $p=0.03$, ES=0.50] compared to the SI and control group. The SA group also showed a trend towards higher stimulated production of IL–6 compared to the SI and control group in the reduced sample (β=0.33, 95% CI –0.04–0.7), $p=0.08$, ES=0.49). |

*(Continued)*

| Author and date | Sample type and demographics | Study design | Country | Exposure: risk or protective factors measured | Suicide measure | Significant risk or protective factors for suicide attempts | Study findings |
|---|---|---|---|---|---|---|---|
| Mortier et al [100]. | 13,984 students (54.4% female), $M_{age}$ =19.33 years, $SD$=0.59. | Cross-sectional | Eight countries: Australia, Belgium, Germany, Mexico, Northern Ireland, South Africa, Spain, US. | Age, gender, parental education, parents not married or at least one parent deceased, place raised, religion, sexual orientation, current living situation, expected to work on student job, self-reported ranking in high school, most important reason to go to college extrinsic. | A modified version of the C-SSRS [53]. SI: "Did you ever wish you were dead or would go to sleep and never wake up?" and "Did you ever in your life have thoughts of killing yourself?". SA: "Have you ever made a suicide attempt [i.e., purposefully hurt yourself with at least some intent to die]?". | Age 20 and older, being female, having been raised in a large city, high parental education, non-heterosexual orientation with same-sex sexual intercourse, and non-heterosexual orientation without same-sex sexual intercourse. | Unplanned attempts among those with lifetime SI were significantly predicted by being 20 or older at matriculation (aOR = 2.5, 95% CI 1.1–5.7) and non-heterosexual orientation with same-sex sexual intercourse (aOR = 6.1, 95% CI 2.5–14.5). Planned attempts among those with SI were significantly predicted by non-heterosexual orientation with same-sex sexual intercourse (aOR = 2.5, 95% CI 1.6–4.0), non-heterosexual orientation without same-sex sexual intercourse, being female (aOR = 1.9, 95% CI 1.1–3.1), having been raised in a large city (aOR = 1.8, 95% CI 1.2–2.8) and by high parental education (compared to medium parental education; aOR = 0.7, 95% CI 0.5–1.0). |
| Musci et al [101]. | 581 6th–10th grade students (47.3% female), aged 11–19 years. | Longitudinal | US | Gender | DISC [102]. Single item measures. SI: "In the last year did you think seriously about killing yourself?" (using responses from sixth to tenth grades). SA: "In the past year have you tried to kill yourself, or made a suicide attempt?" (Measured at ages 19–22 annually). | Female gender | The SI trajectory significantly predicted SA after high school as those experiencing SI had a 0.37 probability of reporting a SA compared to 0.09 for those not experiencing SI ($\chi^2$ (1) = 12.6, $p<0.01$). In this post high school SA, females were significantly more likely to report suicide attempts compared to males ($\chi^2$=10.612, $p$ =.005). |
| Nestor et al [103]. | 4500 adolescents (52.9% female), $M_{age}$ =16.6 years, 11.7% Hispanic origin, 68.9% White, 22.6% African American, 3.9% Native American, and 4.0% Asian American. Wave 1: SI ($N$=413). SA ($N$=124). Wave 2: SI ($N$=313). SA ($N$=153). | Cross-sectional and longitudinal | US | Age, gender, ethnoracial minority, exposure to SA in family, exposure to SA in friends, perceived social support, depression. | Single-item measures: SI: "During the past 12 months, did you ever seriously think about committing suicide?" SA: "During the past 12 months, how many times did you actually attempt suicide?". | Depression (cross-sectional analyses). Age, ethnoracial minority, previous SA, depression (longitudinal analyses). | In the cross-sectional analyses, only depression significantly distinguished the SA group from the SI group (OR = 1.03, 95% CI 1.001–1.050). In the longitudinal analyses, age (OR = 0.82, 95% CI 0.72–0.95), ethnoracial minority (OR = 1.64, 95% CI 1.01–2.69), previous SA (OR = 1.79, 95% CI 1.26–2.55), and depression (OR = 1.04, 95% CI 1.01–1.07) significantly distinguished the SA group from the SI group. |

*(Continued)*

**Table 2.** (Continued)

| Author and date | Sample type and demographics | Study design | Country | Exposure: risk or protective factors measured | Suicide measure | Significant risk or protective factors for suicide attempts | Study findings |
|---|---|---|---|---|---|---|---|
| Okado et al [33]. | 8113 9th-12th grade adolescents (54% female), 25.4% Native Hawaiian/ part Hawaiian, 18.7% Filipino, 17.2% Hispanic/ Latino, 16.2% multi-racial, 8.6% White, 4.4% other Pacific Islander, 4.4% Japanese, 4.2% Other Asian, 0.7% Black/ African American, and 0.2% American Indian/ Alaska Native. $M_{age}$ and $SD$ not reported. | Cross-sectional | Hawaii, US | Disinhibition, sports participation, academic performance. | Three items from the Hawaii Youth Risk Behavior Survey were used to measure SI and SA. SI: Adolescents reported whether they had seriously considered suicide (yes or no). SA: Adolescents reported whether they made plans to attempt suicide and how many times they had attempted suicide (0=no attempt, 1=one or more attempts). | Disinhibition, higher academic performance (protective factor). | Disinhibition interacted with SI to predict greater likelihood of SA ($\beta$=1.68, $p$=.002). Higher academic performance interacted with SI to predict lower likelihood of SA ($\beta$=0.50, $p$=<.001). |
| Ozger et al [104]. | Adolescent inpatients with MDD. SI: $N$=9 (77.78% female), $M_{age}$=15.41 years, $SD$=1.24. SA: $N$=17 (76.47% female), $M_{age}$=15.74 years, $SD$=1.06. | Cross-sectional | US | Anxiety, depression, brain functioning including electrode pairs Fp2-F4 Delta, Fp2-F4 Alpha, Fp2-F4 Beta, C4-P4 Alpha, C4-P4 Beta, T8-P8 Theta, T8-P8 Alpha, T8-P8 Beta, P7-O1 Delta. | C-SSRS [53] assessed current SI and suicidal behaviour. | Electrode pairs: Fp2-F4 Delta, Fp2-F4 Alpha, Fp2-F4 Beta, P7-O1 Delta. | Electrode pairs Fp2-F4 Delta (Mean difference=0.099, SE=0.036, $p$=0.023), Fp2-F4 Alpha (Mean difference=0.088, SE=0.032, $p$=0.024), Fp2-F4 Beta (Mean difference=0.113, SE=0.035, $p$=0.006), and P7-O1 Delta (Mean difference=0.100, SE=0.037, $p$=0.028) significantly distinguished the SA group from the SI group. All other electrode pairs including C4-P4 Alpha (Mean difference=-0.121, SE=-0.121, $p$=0.07), C4-P4 Beta (Mean difference=-0.100, SE=0.047, $p$=0.119), T8-P8 Theta (Mean difference=-0.0019, SE=0.031, $p$=1.00), T8-P8 Alpha (Mean difference=-0.026, SE=0.043, $p$=1.00), and T8-P8 Beta (Mean difference=-0.024, SE=0.038, $p$=1.00) did not significantly distinguish the SA group from the SI group. |

*(Continued)*

**Table 2.** (Continued)

| Author and date | Sample type and demographics | Study design | Country | Exposure: risk or protective factors measured | Suicide measure | Significant risk or protective factors for suicide attempts | Study findings |
|---|---|---|---|---|---|---|---|
| Paul [105] | 928 adolescents aged 13–18 years with lifetime SI from the National Comorbidity Survey-Adolescent Supplement. SI group: $N$ = 847 (60.6% female), $M_{age}$ = 15.52 years, $SD$ = 1.48. SA group: $N$ = 81 (80.2% female), $M_{age}$ = 15.16 years, $SD$ = 1.37. | Cross-sectional | US | Age, gender, race/ethnicity, family income, parental education, poverty index, living with biological parents, parent suicide attempt, lifetime plan, past year DSM-IV (mood, anxiety, disruptive behaviour, and substance use), past year stressful life events (romantic break-up, close-friend break-up, death of close friend/ family, serious illness or injury of close friend, big disappointment, number of past year stressful life events), and past year serious ongoing problems (spouse/ romantic partner, sibling, parent/ close relative, friends, supervisor/ teacher), number of past year problems. | A modified version of the Suicidal Behavior Module of the World Health Organization CIDI [89,91]. | Romantic break-up, greater past year stressful life events. | The SA group were significantly more likely to have a romantic break-up (aOR = 1.88, 95% CI 1.11–3.18) and experience greater past year life events (aOR = 1.52, 95% CI 1.02–1.35) than the SI only group. |

*(Continued)*

| Author and date | Sample type and demographics | Study design | Country | Exposure: risk or protective factors measured | Suicide measure | Significant risk or protective factors for suicide attempts | Study findings |
|---|---|---|---|---|---|---|---|
| Plener et al [106]. | 665 adolescent students (57.0% female), aged 14–17 years, ($M_{age}$ = 14.8, $SD$ = 0.66). SI group: $N$ = 192. SA group: $N$ = 43. $M_{age}$, $SD$, and sex/gender not reported per group. | Cross-sectional | Germany | Traumatic events (Serious accident, fire, or explosion, natural disaster, nonsexual assault by a family member or someone you know, nonsexual assault by a stranger, sexual assault by a family member or someone you know, sexual assault by a stranger, military combat or war zone, sexual contact with person >5 years older, torture, life-threatening illness, and other traumatic events). | German translation of The Self-Harm Behaviour Questionnaire [74]. SI: "Have you ever talked or thought about committing suicide?" SA: "Have you ever attempted suicide?". | Less traumatic events, greater multiple exposure to traumatic events, sexual abuse. | Students with SA reported significantly less traumatic events ($\chi^2$ (2, $N$ = 235) = 4.44, $p$ = .035), yet reported significantly greater multiple exposure to traumatic events ($\chi^2$ (2, $N$ = 235) = 12.21, $p$ = .002) compared to students with SI only. Students with a history of SA reported greater exposure to sexual abuse compared to students with SI $\chi^2$ (2, $N$ = 235) = 7.2, $p$ = .015). However, no significant differences were found for physical abuse ($\chi^2$ (2, $N$ = 235) = 3.4, $p$ = > .05). |
| Puangsri & Ninla-aesong [107] | 137 young adults with MDD/ university students (70.07% female), aged 18–24 years, Median = 20.0 years, $SD$ = 1.30. | Retrospective cohort, case-control | Thailand | White blood cell count, neutrophil %, lymphocyte %, monocyte %, monocyte count, platelet count, neutrophil-to-lymphocyte ratio, monocyte-to-lymphocyte ratio, platelet-to-lymphocyte ratio. | Suicidality severity was assessed by a qualified psychiatrist during an interview using the 8 Questionnaire (Thai-version of a suicidality module of the M.I.N.I): 5.0.0 [108]. | Monocyte count | The SA group with MDD had significantly higher monocyte count, compared to the SI group (Cohen's $d$ = 0.63, $p$ = 0.017). The SA group with MDD tended to have a higher monocyte to lymphocyte ratio than the SI group with MDD but this was not statistically different ($p$ = 0.062, $d$ = 0.51). |

(Continued)

| Author and date | Sample type and demographics | Study design | Country | Exposure: risk or protective factors measured | Suicide measure | Significant risk or protective factors for suicide attempts | Study findings |
|---|---|---|---|---|---|---|---|
| Quevedo et al [109]. | 129 adolescents aged 12–17 years. Depressed adolescents with low SI: $N=33$ (60.60% female), $M_{age}=14.99$ years, $SD=1.76$. Depressed adolescents with high SI: $N=28$ (67.86% female), $M_{age}=15.11$ years, $SD=1.6$. Depressed adolescents with a SA in the past 6 months: $N=28$ (75.0% female), $M_{age}=14.67$ years, $SD=1.57$. Healthy youth: $N=40$ (47.5% female), $M_{age}=14.38$ years, $SD=1.53$. | Cross-sectional | US | Caudate body and head brain results. | K-SADS-PL [43]; the Child Depression Rating Scale [110]. | Lower bilateral caudate activity during positive self-processing, blunted reward circuitry during positive vs. negative self-related material. | The SA group displayed lower bilateral caudate (body and head) activity during positively valenced self-processing compared to the high SI and low SI group. The SA group reported higher caudate activity than the low SI group, but not the high SI group during negative vs. positive self-appraisals engaged the caudate. This activity also extended to thalamus, precuneus, thalamus and posterior cingulate cortex. |
| Rajappa et al [111]. | 96 undergraduates (76.04% female), aged 18–30 years ($M_{age}=19.0$, $SD=2.2$). 31% Asian, 30% White, 23% Hispanic, 7% Black, and 8% other. Single SA group ($N=20$), multiple SA group ($N=17$), SI group ($N=17$), control group ($N=42$). | Cross-sectional | US | Age, meeting diagnostic criteria. | SI: BSS (measures current suicidal ideation) [112]. SA: The Suicidal Behaviour Screening derived from the young adult version of the C-DISC [102]. "Have you ever, in your whole life, tried to kill yourself or made a suicide attempt?". More than one previous SA was classified as multiple attempts. | Older age (single SA group compared to SI group), meeting diagnostic criteria (multiple SA group compared to SI group). | The SA group were significantly older than the SI group (t (92) = 2.94, $p<.05$). Those who reported multiple suicide attempts were significantly more likely to meet diagnostic criteria than those with SI only. |
| Ren et al [113]. | 930 Chinese adolescents (53.5% female) aged 15–21 years, $M_{age}=16.88$ years, $SD=0.55$. 10th grade students from a senior high school. SI group, SA group, No SI or SA group. $N$, $M_{age}$ and $SD$ not reported. | Cross-sectional | China | Fearlessness, pain tolerance, pain sensitivity, suicide preparation, suicide plan, courage, suicide attitude, NSSI, depressive symptoms, and hopelessness. | Single item measures. SI: "Have you had suicide ideation in the past 12 months?" SA: "Have you attempted suicide/ had suicide ideation in the past 12 months?" | NSSI | The only variable which significantly distinguished the SA group from the SI group was NSSI (β = 1.83, Wald=3.91, OR = 6.26, 95% CI 1.02–38.55). |

(Continued)

| Author and date | Sample type and demographics | Study design | Country | Exposure: risk or protective factors measured | Suicide measure | Significant risk or protective factors for suicide attempts | Study findings |
|---|---|---|---|---|---|---|---|
| Rengasamy et al [114]. | Adolescents aged 11–18 years psychiatrically hospitalized within a general adolescent inpatient psychiatric unit for acute SI or acute SA. Test cohort: $N$ = 110. Replication cohort: $N$ = 97. SI group (combined cohort): $N$ = 144 (60% female), $M_{age}$ = 14.64 years. SA group (combined cohort): $N$ = 63 (78% female), $M_{age}$ = 15.02 years. | Cross-sectional and longitudinal | US | Complete blood count measures (White blood count, red blood count, hemoglobin, hematocrit mean corpuscular volume mean corpuscular hemoglobin, mean corpuscular hemoglobin concentration, red cell distribution width, mean platelet volume, platelet count, neutrophil percentage, lymphocyte percentage, monocyte percentage, eosinophil percentage, basophile percentage, absolute neutrophil, absolute lymphocyte, absolute monocyte, absolute eosinophils, absolute basophils, neutrophil-lymphocyte ratio. | Board-certified child and adolescent psychiatrist, using standardized criterion established through the C–CASA [76]. | Lower absolute eosinophil counts, lower eosinophil percentages, lower platelet counts, white blood count. | In the test cohort, the SA group had lower absolute eosinophil counts, (Std. $\beta$=−0.58, 95% CI −0.98, −0.18, $d$=−0.6, $p$=0.005, $N$=107), lower eosinophil percentages (Std. $\beta$=−0.42, 95% CI −0.82, −0.01, $d$=−0.42, $p$=0.044, $N$=107), and lower platelet counts (Std. $\beta$=−0.48, 95% CI −0.88, −0.08, $d$=−0.49, $p$=0.019, $N$=109). However, platelet levels were no longer significant when controlling for duration between medical stabilization and complete blood count measures. In the replication cohort, the SA group (relative to the SI group) had lower eosinophil percentages (Std. $\beta$=−0.5, 95% CI −0.96, −0.04, $d$=−0.51, $p$=0.035, $N$=88). However, in the replication cohort, no significant associations emerged for absolute eosinophil count (Std. $\beta$=−0.11, 95% CI −0.59, 0.37, $d$=−0.11, $p$=0.647, $N$=87). In contrast to the test cohort, the SA group had significantly greater platelet counts (Std. $\beta$=0.45, 95% CI 0.01, 0.89, $d$=0.46, $p$=0.044, $N$=96). In longitudinal analyses, adolescents with SI at baseline, both lower eosinophil percentages (Std. $\beta$=−0.74, 95% CI −1.3, −0.18, Cohens $f^2$=0.052, $p$=0.01, $N$=136) and lower eosinophil counts (Std. $\beta$=−0.85, 95% CI −1.41, −0.29, Cohens $f^2$=0.068, $p$=0.003, $N$=136) were associated with prospective suicidal behaviour. In the combined cohort (test and replication), the SA group had both lower eosinophil percentages (Std. $\beta$=−0.39, 95% CI −0.69, −0.08, Cohens $f^2$=0.032, $p$=0.014, $N$=195) and lower absolute eosinophil counts (Std. $\beta$=−0.36, 95% CI −0.67, −0.05, Cohens $f^2$=0.028, $p$=0.023, $N$=194), adjusting for age and gender. Platelet counts were not associated with SA ($p$=0.87, $N$=205). For the combined cohort, the SA group had lower eosinophil counts than the SI group. The SA group also had a higher white blood count than the SI group (Std. $\beta$=0.37, $p$=0.016). No other effects were significant. |

*(Continued)*

**Table 2.** (Continued)

| Author and date | Sample type and demographics | Study design | Country | Exposure: risk or protective factors measured | Suicide measure | Significant risk or protective factors for suicide attempts | Study findings |
|---|---|---|---|---|---|---|---|
| Rengasamy et al [115]. | 76 psychiatric inpatients (42.9% female), aged 15–30 years, $M_{age}$ =22.87 years, $SD$=3.4. SI with no prior SA: $N$=38 (55% female), $M_{age}$ =23.58 years, $SD$=3.8. SA group: $N$=38 (71% female), $M_{age}$ =22.79 years, $SD$=3.84. Healthy controls: $N$=36 (44% female), $M_{age}$ =22.19 years, $SD$=2.23. | Cross-sectional | US | Cytokine pathway markers (e.g., cytokines and proteins in cytokine signalling pathways). | Patients admitted for SA or SI. SI severity was assessed by the ASIQ [116]. | No factors were significant. | There were no significant differences between SI and SA groups in cytokine pathway markers. |
| Rivers et al [117]. | 4,233 students with a history of suicide risk (60.7% female), $M_{age}$ =14.65 years, $SD$=2.06, 59.8% White, 16.0% Black, 15.4% multiracial, 8.8% other, and 14.4% Hispanic. SA group: $N$=1437 (68.3% female), $M_{age}$ =14.76 years, 55.5% White. SI group: $N$=2054 (56.2% female), $M_{age}$ =14.58 years, 62.3% White. | Cross-sectional | US | Age, gender, race, school factors (skips classes often, grades declining, verbal bullying, cyberbullying, no friends at school), factors at home (frequent violence at home, neighbourhood violence, access to gun, parents highly critical, frequent arguing in home, no adults support, parents never know location) and mental health (anxiety, depression, traumatic distress, eating disorder, and substance use). | Three dichotomous items from the Behavioural Health Screen [79] was used to categorize three groups: SI, suicidal plan, and SA groups. | Age, female gender, black ethnicity, school factors (skips classes often and cyberbullying), home factors (access to gun), and mental health (depression, traumatic distress, eating disorder, and substance use). | Relative to the SI group, the SA group were significantly more likely to be older (OR = 1.05, 95% CI 1.01, 1.09), female (OR = 1.63, 95% CI 1.39, 1.92) and of Black ethnicity (OR =1.32, 95% CI 1.07, 1.62). Relative to the SI group, the SA group were significantly more likely to skip school (OR = 1.37, 95% CI 1.06, 1.77), be cyberbullied (OR = 1.82, 95% CI 1.14, 2.93), or have access to a gun (OR = 1.46, 95% CI 1.13, 1.90). Relative to the SI group, the SA group reported significantly greater depression (OR = 1.14, 95% CI 1.02, 1.26), traumatic stress (OR = 1.39, 95% CI 1.16, 1.66), eating disorder (OR = 1.46, 95% CI 1.16, 1.84), and substance use (OR = 1.79, 95% CI 1.38, 2.34). |

*(Continued)*

| Author and date | Sample type and demographics | Study design | Country | Exposure: risk or protective factors measured | Suicide measure | Significant risk or protective factors for suicide attempts | Study findings |
|---|---|---|---|---|---|---|---|
| Robinson et al [118]. | Adolescents aged 13–18 years. SI group: $N$=642 (65.2% female, 34.2% male, 0.6% transgender/ gender diverse), $M_{age}$ = 15.65 years, $SD$ = 1.20. SA group: $N$=82 (68.3% female, 29.3% male, 2.4% transgender/ gender diverse), $M_{age}$ = 15.79 years, $SD$ = 1.27. | Cross-sectional | New Zealand | NSSI thoughts and NSSI | SBQ-R [39]. | NSSI | The SA group reported significantly greater NSSI compared to the SI group (OR = 7.51, 95% CI 3.48, 16.24). |
| Rooney et al [119]. | 821 youth aged 13–21 years (63.1% female), $M_{age}$ = 16.20 years, $SD$=1.66, 64.7% White, 16.8% African American/ Black, 3.9% Asian, 1.7% Native American, 6.4% other, 6.5% biracial and 12.1% Hispanic. | Cross-sectional | US | Age, gender, ethnicity, income, binge drinking, marijuana use, non-violent delinquency, depression, violence perpetration, violence victimization and perpetration x victimization. | SA measured using one item that assessed the frequency with which youth attempted suicide in the last 12 months. | Lower income, black youth (Step 1), non-violent delinquency, depressive symptoms, (Step 2), more frequent victimization experiences (Step 3), interaction of violence perpetration by victimization (Step 4). | Lower income and being Black was associated with greater SA among youth with SI (Step 1). Greater endorsement of non-violent delinquency and depressive symptoms were associated greater SA among youth with SI (Step 2). More frequent victimization experiences were associated with greater SA among youth with SI (Step 3). The interaction of violence perpetration by victimization was associated with greater SA among youth with SI (Step 4). |
| Saffer et al [120]. | Adolescent psychiatric inpatients: $N$=172, (77% female), $M_{age}$ = 15.04 years, $SD$ = 1.39, 68% Caucasian, 10% African American, 21% Hispanic, and 1% Asian. High school students: $N$=426 (61% female), 13–17 years, 53% Caucasian, 19% Hispanic, 15% Asian, 11% African American, and 3% mixed race. | Retrospective, Cross-sectional | US | Parental bonding, emotion dysregulation, loneliness, and self-worth. | Items from the Youth Risk Behaviour Survey. SI: "Have you ever seriously thought about killing yourself?" SA: "Have you ever tried to kill yourself?". | Lower parental care (psychiatric sample only). | In the psychiatric sample, lower parental (maternal and paternal) care significantly distinguished adolescents with previous history of SA from those with SI only ($r_{pb}$ (83) = −.47, $p$ <.001 and $r_{pb}$ (75) = −.24, $p$ <.05) respectively, controlling for emotional dysregulation, loneliness, and low self-worth. No significant findings emerged in high school students. |

(Continued)

| Author and date | Sample type and demographics | Study design | Country | Exposure: risk or protective factors measured | Suicide measure | Significant risk or protective factors for suicide attempts | Study findings |
|---|---|---|---|---|---|---|---|
| Santana et al [121]. | 5,037 young adults, aged >18 years, $M_{age}$ = 24.78, $SD$ = 11.65. | Cross-sectional | Brazil | Depression, panic, generalized anxiety disorder, substance use, antisocial personality disorder, number of parental disorders. | The suicidality module of the WMH-CIDI [89]. Examined history of SA and SI. | Panic | Only panic was associated with distinguishing the SA group from the SI group (aOR = 2.7, 95% CI 1.2–5.8). |
| Schlagbaum et al [122]. | 118 adolescents, aged 13–18 years (72.9% female; $M_{age}$ = 15.41, $SD$ = 1.39; 67.8% White, non-Hispanic), were recruited from the emergency department and outpatient mental health clinics. SI group: $N$ = 19 (68.4% female), $M_{age}$ = 15.05 years, $SD$ = 1.35, 63.2% White. SA group: $N$ = 40 (75.0% female), $M_{age}$ = 15.53 years, $SD$ = 1.36, 70% White. | Cross-sectional | US | Age, IQ, depressive symptoms, anxiety symptoms, aggression, peer behaviour, affiliation with peers with suicidal thoughts/behaviors. | Columbia University Suicide History Form [123]. Examined history of SI and SA. | No factors were significant. | There were no statistically significant differences between the SA group and the SI on any risk or protective factors. |
| Scott et al [124]. | 1950 female adolescents from a community sample aged 10–21 years. $M_{age}$ and $SD$ not reported. 55.7% African American, 38.8% Caucasian, 4.9% multiracial, and 0.6% Asian. SI group: $N$ = 609 SI and NSSI group: $N$ = 170. No SI or NSSI group: $N$ = 1171. | Longitudinal | US | NSSI | SI in the past year: Child Symptom Inventory–4th edition used at age 10 (i.e., "Have you thought about death or suicide?"), Adolescent Symptom Inventory–4th edition used at age 12 and the Adult Self-Report Inventory-4 at age 18. SA: SBQ-R (at age 17–21 years). | NSSI | The SI group were significantly less likely to report a lifetime SA, than the SI and NSSI group (β = −1.43, SE = .31, Wald χ2 [1] = 21.69, $p$ < .001, OR = 0.24, 95% CI.13–.44). The SI group were also significantly less likely to report a recent SA than those in the SI and NSSI group (β = 0.88, SE = .42, Wald χ2 [1] = 4.27, $p$ = .04, OR = 0.42, 95% CI.18–.96). When controlling for SI severity, the SI group remained significantly less likely to report a lifetime SA (β = −1.23, SE = .32, Wald χ2 [1] = 14.51, $p$ = .001, OR = 0.29, 95% CI.16–.55) than the SI and NSSI group. When controlling for SI severity (the model no longer predicted SA), and the SI group were marginally less likely to predict recent SA (β = −0.82, SE = .44, Wald χ2 [1] = 3.39, $p$ = .07, OR = 0.44, 95% CI.19–1.05), than the SI and NSSI group. |

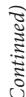

*(Continued)*

| Author and date | Sample type and demographics | Study design | Country | Exposure: risk or protective factors measured | Suicide measure | Significant risk or protective factors for suicide attempts | Study findings |
|---|---|---|---|---|---|---|---|
| Sewall et al [125]. | 386 participants recruited primarily from outpatient clinics (47% female), $M_{age}$ = 14.5 years, 82% White (at baseline). The SI group investigated ($N$=265) was examined for transition to suicide attempts. | Longitudinal | US | Age, sex, race, family, SES family living together, current psychopathology (internalizing disorder, externalizing disorder, hypomania/ mania, thought disorder), family history of psychopathology (internalizing disorder, externalizing disorder, bipolar, thought disorder, suicide) relationship quality (recent relationship quality with parents, recent relationship quality with friends, average relationship quality with parents, average relationship quality with friends) and various interactions of relationship quality. | Weekly SI and SA was measured using the Longitudinal Interval Follow-up Evaluation Self-Injurious/ Suicidal Behavior Scale. | Worsening recent relationship quality with parents. | A three-month period of or poorer relationship quality with parents, relative to their baseline level of relationship quality, was associated with greater likelihood of a suicide attempt (median β= − .15, 95% highest density interval −0.31, 0.01). |
| Soylu et al [126]. | 63 adolescents reporting SI aged 12–18 years (76.19% female), $M_{age}$ =16.01 years, $SD$=1.31, 82.70% White. | Cross-sectional | Turkey | Depression, state anxiety, trait anxiety, hopelessness, self-esteem, perceived social support (family/ friend/ a special person), strengths and difficulties (attention deficit and hyperactivity, behavioural problems, emotional problems, peer problems and prosocial behaviour). | Suicidal Behaviour Assessment Form (developed for this study). Severity of suicidal behaviour was assessed with a scale between 0 and 5: (0=none, 1=only being bored of life, 2=will to die, 3=SI, 4=suicide plan, 5=SA). | Low family support | The SA group reported significantly lower family support, compared to the SI group ($t$=−3.10, $p$=0.003). |

*(Continued)*

| Author and date | Sample type and demographics | Study design | Country | Exposure: risk or protective factors measured | Suicide measure | Significant risk or protective factors for suicide attempts | Study findings |
|---|---|---|---|---|---|---|---|
| Stack [127] | 2536 youth, sex/gender, $M_{age}$ and $SD$ not reported. SI group: $N=1,439$, SA group: $N=1,097$, Sex/ gender, $M_{age}$ and $SD$ not reported. | Cross-sectional | US | Age, gender, race, region of residence, urbanity, violence (fighting, dating violence, dating violence, and weapon carrying), major depression, eating disorder, marijuana use, cocaine use, binge drinking, suicide plan, risky sexual behaviour (number of sexual partners, unsafe sexual intercourse, pregnant, impregnated someone), school integration (sports team member and grade point average). | Single item measures. SI: "During the past 12 months, did you ever seriously consider attempting suicide?". SA: "During the past 12 months, how many times did you actually attempt suicide?" | Being female, younger age, residence, Hispanic, African American, or Asian ethnicity, violence (fighting, dating violence, and weapon-carrying), major depression, eating disorders, cocaine use, having a suicide plan, number of sexual partners. | Demographic factors that distinguished the SA group from the SI group included being female (male, OR = 0.60), age (older age OR = 0.84), residence in the Midwest (OR = 0.61), and Hispanic (OR = 2.15), African American (OR = 1.47), Asian ethnicity (OR = 1.81), or other non-white race (OR = 1.81). Controlling for the other variables, the SA group reported significantly greater fighting (OR = 2.18) and weapon carrying (OR = 1.13), major depression (OR = 1.86), eating disorder (OR = 1.69), use of cocaine (OR = 2.34), having a suicide plan (OR = 2.70), and number of sexual partners (OR = 1.08) compared to the SI group. |
| Stange et al [128]. | 212 young adults. SI group: $N=60$. SA group: $N=18$. Major depression but no suicidal behaviour group: $N=52$. Healthy controls: $N=82$. | Cross-sectional (prospective data is available for a subset of participants). | US | Resting-state intrinsic network connectivity. | Diagnostic Interview for Genetics Studies [129] or the Structured Clinical Interview for DSM-5 [130]. Examined history of SI and SA. | Greater connectivity between the right precuneus and the Salience Emotional Network seeds, and greater connectivity between the middle/ inferior frontal gyrus and the Salience Emotional Network seeds. | The SA group had significantly greater connectivity between the right precuneus and the Salience Emotional Network seeds than the SI group ($p<0.01$). The SA group had significantly greater connectivity between the right middle/superior frontal gyrus and the Salience Emotional Network seeds than the SI group ($p=0.04$). The SA group had significantly greater connectivity between the right middle/ inferior frontal gyrus and the Salience Emotional Network seeds than the SI group ($p=0.03$). |

*(Continued)*

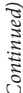

**Table 2.** (Continued)

| Author and date | Sample type and demographics | Study design | Country | Exposure: risk or protective factors measured | Suicide measure | Significant risk or protective factors for suicide attempts | Study findings |
|---|---|---|---|---|---|---|---|
| Stewart et al [12]. | Adolescent inpatient self-injurers (N=397, 79.85% female), 13–18 years, $M_{age}$ = 15.44 years, $SD$ = 1.36. SI group: N=149 (79.73% female), $M_{age}$ = 15.50 years, $SD$ = 1.30. SA group: N=152 (83.78% female), $M_{age}$ = 15.26 years, $SD$ = 1.39. No SI/SA group: N=96 (78.12% female), $M_{age}$ = 15.60 years, $SD$ = 1.38. | Cross-sectional | US | Unipolar mood, social phobia, any substance use, psychotic symptoms, physical abuse, number of disorders, past month SI, past month plans, number of NSSI methods, age of onset NSSI, depressive symptoms, anxiety symptoms, and risky behaviours. | SITBI [45]; BSS (measures current suicidal ideation) [41]. | Greater NSSI methods, psychotic symptoms, engagement in risky behaviour. | The SA group used more NSSI methods (OR = 1.25, 95% CI, 1.04–1.50) and reported greater psychotic symptoms (OR = 5.84, 95% CI 1.57–21.74), and risky behaviour engagement (OR = 1.05, 95% CI 1.01–1.11) compared to the SI group. |
| Stewart et al. [131]. | 99 adolescents (71.72% female) aged 13–18 years, $M_{age}$ = 15.53 years, $SD$ = 1.34, 77.8% White. SI group: N=60 (68.3% female), $M_{age}$ = 15.52 years, $SD$ = 1.32, 76.7% White. Single attempt group: N=12 (66.7% female), $M_{age}$ = 15.58 years, $SD$ = 1.44, 75.0% White. Multiple attempt group: N=26 (80.8% female), $M_{age}$ = 15.58 years, $SD$ = 1.36, 80.8% White. | Cross-sectional | US | Age, sex, psychiatric disorders, deficits in cognitive control in the context of suicide-related stimuli using the Suicide Stroop Task (positive/ negative suicide interference, emotional stimuli, Suicide Stroop Task errors and total latency). | SITBI [45]. | Deficits in cognitive control in the context of suicide-related stimuli using the Suicide Stroop Task, greater interference for suicide and positive stimuli, greater interference for emotional stimuli, regardless of valence. | The SA group had greater interference for suicide ($t_{97}$ = 2.04, $P$ = 0.44, $d$ = 0.41), in comparison to the SI group. Divided into multiple and single SA groups, the multiple SA group had significantly greater interference than the SI group ($p$ = .004, $d$ = .004), however the single SA group and the SI group did not significantly differ ($p$ = .629, $d$ = −0.12). The SA group and the SI group did not significantly differ in negative interference scores ($t_{97}$ = 0.60, $P$ = .549, $d$ = 0.12). The SA group had greater interference for positive words/ stimuli ($t_{97}$ = 2.63, $P$ = 0.10, $d$ = 0.53). Divided into multiple and single SA groups, the multiple SA group had significantly greater positive interference than the SI group ($p$ = <.001, $d$ = 0.79), however the single SA group and the SI group did not significantly differ ($p$ = .823, $d$ = −0.05). The SA group had greater interference for emotional stimuli, regardless of valence. In addition, the SA group and SI group did not significantly differ in the number of Suicide Stroop Task errors or in total reaction time across valences. |

**Table 2.** (Continued)

| Author and date | Sample type and demographics | Study design | Country | Exposure: risk or protective factors measured | Suicide measure | Significant risk or protective factors for suicide attempts | Study findings |
|---|---|---|---|---|---|---|---|
| Stewart et al [132]. | 197 adolescent psychiatric patients (73.10% female), aged 13–19 years, $M_{age}$ =15.61 years, $SD$ =1.48. SI group: $N$=99. SA group: $N$=60. No SI/SA group: $N$=38. 81.22% White, 10.15% multicultural, 5.58% Asian, 2.03% Black, and 0.51% Native Hawaiian or Pacific Islander. | Cross-sectional, case control. | US | Abuse, mood disorder, anxiety disorder, number of disorders, depressive symptoms, hopelessness, SI. Life stress/ events and chronic difficulties in five social-psychological domains: Interpersonal loss, physical danger, humiliation, entrapment, and role change/ disruption. | SITBI [45]; BSS (measures current suicidal ideation) [41]. | Interpersonal loss, physical and sexual abuse, less severe depressive symptoms. | The SA group reported higher rates of physical and sexual abuse (OR = 2.79, 95% CI 1.48–5.26) and less severe depressive symptoms (OR = 0.95, 95% CI 0.93–0.97) compared to the SI group. For life stress/ events and chronic difficulties, only interpersonal loss events distinguished the SA group from the SI group (OR = 1.49, 95% CI 1.15–1.94). The SA group were significantly more likely to report interpersonal loss events that occurred within two months of hospitalization (OR = 1.85, 95% CI 1.30–2.64), compared to the SI group. When restricting the SA group to those with a single attempt only, this finding remained significant. There were no significant differences in endorsing any mood disorder and anxiety disorders between the SA and SI groups. |

*(Continued)*

| Author and date | Sample type and demographics | Study design | Country | Exposure: risk or protective factors measured | Suicide measure | Significant risk or protective factors for suicide attempts | Study findings |
|---|---|---|---|---|---|---|---|
| Taliaferro & Muehlenkamp [133]. | 35339 9th and 12th grade students (50% female), $M_{age}$, $SD$=not reported. 76.9% White, 4.7% African/ American 5.4% Asian/ Pacific Islander, 3.8% Hispanic, 1.1% Native American, 6.5% mixed race, and 1.7% 'don't know'. | Cross-sectional | US | Ethnicity, grade, free lunch, living with biological parents, victim/ perpetrator of social-verbal bullying, family substance use, witness to family violence, physical abuse, sexual abuse, victim of dating violence, mental health and physical health problems, cigarette smoking, alcohol use, binge drinking, marijuana use, misuse of prescription drugs or use of illegal drugs, ran away from home, skipped school because felt unsafe, NSSI, perpetrator of violence, victim of school violence, perceived over-weight/ maladaptive dieting behaviour, same-sex sexual experience, depressive symptoms, hopelessness, stress/ anxiety, and distractibility/ impulsivity. | Single item measures. SI: "Have you ever thought about killing yourself?" SA: "Have you ever tried to kill yourself?" | NSSI, a mental health problem, dating violence victimization, and running away from home, cigarette smoking (males only), being white (females only), physical and sexual abuse (females only), same-sex sexual experience (females only), hopelessness (females only), stress or anxiety (females only). Protective factors: parental connectedness, GPA, academic achievement, neighbourhood safety, being white (males only), living with both biological parents (females only). | Factors that significantly distinguished the SA group from the SI group included dating violence victimization (male: OR = 1.51, 95% CI 1.07–2.12; female: OR = 1.58, 95% CI 1.25–1.99), a mental health problem (male: OR = 1.85, 95% CI 1.38–2.48; female: OR = 2.30, 95% CI 1.82–2.91), running away from home (male: OR = 2.00, 95% CI 1.47–2.73; female: OR = 1.80, 95% CI 1.42–2.28), and NSSI (male: OR = 5.17, 95% CI 3.93–6.81; female: OR = 3.90, 95% CI 3.05–4.98). Among males only, cigarette smoking significantly distinguished the SA group from the SI group (OR = 1.52, 95% CI 1.07–2.15). Among females, being white (OR = 0.66, 95% CI 0.52–0.84), physical abuse (OR = 1.66, 95% CI 1.28–2.16), sexual abuse (OR = 1.43, 95% CI 1.12–1.82), a same-sex sexual experience (OR = 2.12, 95% CI 1.42–3.16), greater hopelessness (OR = 1.95, 95% CI 1.21–3.14), and stress or anxiety (OR = 0.50, 95% CI 0.27–0.91) significantly distinguished the SA group from the SI group. Protective factors that distinguished the SA group from the SI group were parent connectedness (male: OR = 0.41, 95% CI 0.27–0.72; female: OR = 0.28, 95% CI 0.18–0.45), academic achievement (male: OR = 0.49, 95% CI 0.31–0.78; female: OR = 0.30, 95% CI 0.20–0.44), and perceptions of neighbourhood safety (male OR = 0.35, 95% CI 0.19–0.67; female OR = 0.44, 95% CI 0.24–0.81). Among males only, being white (OR = 0.76, 95% CI 0.59–0.96) and among females only living with biological parents (OR = 0.79, 95% CI 0.65–0.95) significantly distinguished the SA group from the SI group. |

*(Continued)*

| Author and date | Sample type and demographics | Study design | Country | Exposure: risk or protective factors measured | Suicide measure | Significant risk or protective factors for suicide attempts | Study findings |
|---|---|---|---|---|---|---|---|
| Tanner et al [134]. | 77 adolescents SI group: $N=34$ (73.5% female), $M_{age}=16.15$ years, $SD=0.74$. SA group: $N=15$ (73.3% female), $M_{age}=15.73$ years, $SD=0.88$. No SI/SA group: $N=28$ (57.1% female), $M_{age}=16.07$ years, $SD=0.98$. Ethnicity not reported. | Cross-sectional | Western Australia | Behavioural inhibition and activation, negative life events (e.g., bullying, physical abuse), psychological distress, problem-solving coping, survival and coping beliefs, NSSI, physical abuse, exposure to suicide of a friend or family member. | Single item measures. SI: "Have you ever thought about ending your life?". SA: "Did you ever try and end your life?". | More severe NSSI, exposure to suicide of a family or friend, physical abuse, lower survival and coping beliefs. | Adolescents reporting a prior SA engaged in more severe NSSI compared to adolescents reporting SI ($p<.001$). Adolescents reporting a SA reported lower survival and coping beliefs than adolescents reporting SI ($p<.001$). Adolescents reporting a SA experienced greater serious physical abuse ($\chi^2=5.25$, OR = 4.41, 95% CI 1.19–16.36) and exposure to suicide of a family or friend ($\chi^2=5.55$, or = 1.49–20.72). No significant differences on bullying victimization and sexual assault were found between those reporting SA and those reporting SI. No differences in behavioural inhibition and activation, negative life events, psychological distress, or problem-solving coping were reported between adolescents with a prior SA and those experiencing SI ($p>.05$). |
| Valderrama et al. [135]. | 426 undergraduates (78% female), aged 18–43 years ($M_{age}=19.30$ years, $SD=2.91$). 38% Asian, 24% White, 19% Hispanic/ Latino/a, 9% Black, and 9% Other. SI group: $N=48$ SA only group: $N=25$ SI and SA group: $N=11$. No SI or SA group: $N=342$. | Cross-sectional | US | Brooding, reflection, impulsivity dimensions: lack of premeditation, negative urgency, sensation-seeking, lack of perseverance, general trauma, physical punishment, emotional abuse, sexual abuse, early life trauma (total). | Suicidal Behavior Screening adapted from the young adult version of the C-DISC [102]. SI: "In the past 6 months, have you thought about killing yourself?". SA: "Have you ever, in your whole life, tried to kill yourself or made a suicide attempt?". | Early life trauma, greater sexual abuse. | The SI and SA group had significantly higher total early life trauma scores than the SI only group, but not significantly higher than the SA only group ($F=17.32$, $p<.01$). The SI and SA group had significantly greater sexual abuse scores than the SI only group ($F=14.92$, $p<.01$). The SA only group reported significantly lower lack of perseverance than the SI group. |
| Vélez-Grau et al. [136]. | 14,956 adolescents in 144 schools in 2017 and 13,872 adolescents in 136 schools in 2019, (49.64% female) aged 14–18 years. $M_{age}$ and $SD$ not reported. | Cross-sectional | US | Suicide capability, carried a weapon, carried a weapon at school, carried a gun, fighting, fighting at school, sexual violence, sexual dating violence, physical dating violence, injection drug use, and concussion. | Single item measures. SI: "During the past 12 months, did you ever seriously consider attempting suicide?" SA: "During the past 12 months, how many times did you actually attempt suicide?". | Dual suicide ideation (suicidal thoughts and a plan), suicide capability (carried a weapon, carried a gun, fighting, sexual violence, sexual dating violence, physical dating violence, injection drug use, concussion). | The relationship between SI and SA depended on participants' levels of suicide capability (capability×single ideation: aOR = 0.65, 95% CI 0.43, 0.98, $p=0.041$; capability×dual ideation: aOR = 0.55, 95% CI 0.37, 0.81, $p=0.003$). |

| Author and date | Sample type and demographics | Study design | Country | Exposure: risk or protective factors measured | Suicide measure | Significant risk or protective factors for suicide attempts | Study findings |
|---|---|---|---|---|---|---|---|
| Vergara et al. [137]. | 223 adolescent inpatients who had endorsed past year NSSI (78.9% female), aged 13–18 years ($M_{age}$ = 15.31 years, $SD$ = 1.34), 79.4% White. SI group (endorsed past year NSSI): $N$ = 106 (81.70% female), $M_{age}$ = 15.48 years, $SD$ = 1.36. SA group (endorsed past year NSSI): $N$ = 117 (82.70% female), $M_{age}$ = 15.16 years, $SD$ = 1.30. | Cross-sectional | US | Age, sex, ethnicity, peer victimization, bully perpetration, number of NSSI methods. | SITBI [45]; BSS (measures current suicidal ideation) [41]. | Peer victimization, bully perpetration greater number of NSSI methods. | At the univariate level, both peer victimization ($t$ (220) = 2.73, $p$ = 0.007) and bully victimization ($t$ (212) = 2.69, $p$ = 0.008), significantly distinguished the SA and SI groups. Among the SA group, bully perpetration was associated with a greater number of past attempts ($b$ = 0.07, RR = 1.07, $p$ = 0.02). Being in the SA group was significantly associated with using a greater number of NSSI methods ($p$ < 0.001), compared to the SI group. |
| Voss et al [138]. | 1180 adolescents and young adults (48.3% female), aged 14–21 years, $M_{age}$ = 17.9 years, $SD$ = 2.3. Ethnicity not reported. | Cross-sectional | Germany | Sex, age of onset of SI. | Fully standardized computer-assisted Munich – CIDI [139]. SI: "Have you ever thought over a period of days or weeks about killing yourself, i.e., to attempt suicide?". SA: "Have you ever attempted suicide?". | No factors were significant. | No significant sex differences were associated in the transition from SI to SA (HR = 2.14, 95% CI 0.92–4.95). No associations were found between age at onset of SI (HR = 0.95, 95% CI 0.77–1.16) and the transition to a SA. |
| Wang et al. [140]. | 12,487 9th–12th grade students, aged approx. 14–18 years. SI group: $N$ = 2428 (66.19% female) SA group: $N$ = 961 (65.56% female). | Cross-sectional | US | Alcohol use, cigarette smoking. Covariates: age, sex, grade, ethnicity, academic performance, feeling sad or hopeless. | Single-item measures: SI: "During the past 12 months, did you ever seriously consider attempting suicide?". SA: "During the past 12 months, how many times did you actually attempt suicide?". | Cigarette smoking, lower grades (9th and 10th grade). Greater academic achievement (protective factor). | Cigarette smoking significantly distinguished SA from SI (OR = 1.40, 95% CI 1.10–1.78). Students with higher academic performance, including grades of mostly A's (OR = 0.58, 95% CI 0.44–0.77) and mostly B's (OR = 0.66, 95% 0.51–0.87) were significantly less likely to be in the SA group compared to the SI group. Being in the 9th (OR = 2.33, 95% CI 1.28–4.24) or 10th grade (OR = 2.13, 95% CI 1.28–3.55) significantly distinguished SA from SI. However, age, being female (OR = 1.22, 95% CI 0.97–1.53), and being Hispanic/Latino (OR = 1.10, 95% CI 0.85–1.43) did not significantly distinguish SA from SI. In addition feeling sad or hopeless (OR = 1.25, 95% CI 0.96–1.25) and alcohol use did not significantly distinguish SA from SI (OR = 1.05, 95% CI 0.82–1.35). |

*(Continued)*

| Author and date | Sample type and demographics | Study design | Country | Exposure: risk or protective factors measured | Suicide measure | Significant risk or protective factors for suicide attempts | Study findings |
|---|---|---|---|---|---|---|---|
| Wang et al [141]. | Adolescents diagnosed with depression. SI group: $N$=1138 (75.7% female), $M_{age}$=15.1 years, $SD$=1.6. SA group: $N$=1016, 83.0% female; $M_{age}$=14.8 years, $SD$=1.7. | Cross-sectional | China | Age, sex, total years of education, nation, BMI, residence, annual family income, living arrangement, schooling status, whether an only child, education level of father, education level of mother, marital status of parents, perceived parental relationship, left-behind experience, class ranking, changes in academic performance, academic stress, physical diseases, and smoking status. | Response options: no suicidal thoughts or behaviors, past suicidal thoughts but no suicidal behaviors, and past suicidal behaviors. | Schooling status, total years of education, and loneliness. | Factors related to schooling status (SI=3.5%, SA=10.1%), total years of education (SI=2.6%, SA=8.6%), and loneliness (SI=2.3%, SA=7.4%) were more closely associated with SA relative to SI. Academic stress (SI=7.2%, SA=1.6%), hopelessness (SI=9.1%, SA=5.0%), and age (SI=7.1%, SA=3.2%) were more closely associated with SI than SA. |
| Wang et al [142]. | 17209 9th–12th grade students aged approx. 14–18 years. SI group: $N$=3679 (66.87% female) SA group: $N$=1904 (63.40% female). | Cross-sectional | US | Age of alcohol use initiation (13 years or older, or before age 13). Covariates: age, sex, grade, ethnicity, feeling sad or hopeless, poor mental health, ≥8 hours of sleep, cigarette smoking, electronic vapour product, marijuana use. | Single-item measures: SI: "During the past 12 months, did you ever seriously consider attempting suicide?". SA: "During the past 12 months, how many times did you actually attempt suicide?". | Alcohol use before age 13, marijuana use, cigarette smoking, Hispanic/Latino ethnicity, being in the 9th or 10th grade, poor mental health. Being aged ≤14 years, feeling sad or hopeless (protective factors). | Relative to the SI group, students who drank alcohol before age 13 years were more likely to attempt suicide (OR = 1.61, 95% CI 1.27–2.05). Marijuana use (OR = 1.75, 95% CI 1.42–2.16) and cigarette smoking (OR = 1.67, 95% CI 1.35–2.06) were significantly greater in the SA group than the SI group. Students aged ≤14 years (OR = 0.46, 95% CI 0.24–0.88) were significantly less likely to be in the SA group than the SI group. Students who were Hispanic or Latino (OR = 1.23, 95% CI 1.00–1.51) and in the 9th (OR = 3.26, 95% CI 1.89–5.61) or 10th grade (OR = 1.92, 95% CI 1.23–3.01) were significantly more likely to be in SA group than the SI group. Students who felt sad or hopeless were less likely to be in the SA group relative to the SI group (OR = 0.78, 95% CI 0.63–0.97) and students with poor mental health were more likely to be in the SA group relative to the SI group (OR = 1.21, 95% CI 1.02–1.44). No other variables significantly distinguished the SA and SI groups. |

*(Continued)*

| Author and date | Sample type and demographics | Study design | Country | Exposure: risk or protective factors measured | Suicide measure | Significant risk or protective factors for suicide attempts | Study findings |
|---|---|---|---|---|---|---|---|
| Wetherall et al [143]. | 3435 young people (49.4% female) aged 18–34 years, $M_{age}$ = 25.70, $SD$ = 4.86), 93.80% White. SI group: 45.1% female; $M_{age}$ = 25.33 years, $SD$ = 4.70, 93.5% White. SA group: 60.5% female; $M_{age}$ = 26.54 years, $SD$ = 4.70, 95.1% White. | Cross-sectional | Scotland, UK | Age, gender, ethnicity, marital status, economic activity, employed, economically inactive, unemployed, exposure to suicide death (family and friend), exposure to suicide attempt (family), exposure to suicide attempt (friend), depressive symptoms, defeat, entrapment, burdensomeness, belongingness, goal disengagement, goal reengagement, social support, resilience, acquired capability, mental images, and impulsivity. | Items drawn from the Adult Psychiatric Morbidity Survey. SI (lifetime history): "Have you ever seriously thought of taking your life, but not actually attempted to do so?". SA (lifetime history): "Have you ever made an attempt to take your life, by taking an overdose of tablets or in some other way?". | Older, age, female, higher acquired capability, mental imagery about death, impulsivity, exposure to a suicide attempt of a friend. | The SA group were significantly more likely to be of an older age (aOR = 1.07, 95% CI 1.03–1.10) and female (aOR = 0.49, 95% CI 0.36–0.67), compared to the SI group. The SA group also reported significantly higher levels of acquired capability (aOR = 1.10, 95% CI 1.06–1.14), impulsivity (aOR = 1.02, 95% CI 1.01–1.04), mental images about death (aOR = 1.07, 95% CI 1.03–1.10), and exposure to a suicide attempt of a friend (aOR = 1.49, 95% CI 1.09–2.06), compared to the SI group. No other variables examined differed significantly between the SA and SI group. |
| Yan et al. [144]. | 30,644 Chinese students (43.3% female), $M_{age}$ = 14.14 years, $SD$ = 1.46. SI group: $N$ = 3956 (59.6% female), $M_{age}$ = 15.21, $SD$ = 1.38. SA group: $N$ = 1339 (38.7% female), $M_{age}$ = 15.25 years, $SD$ = 1.48. No SI or SA group: $N$ = 25349 (40.9% female), $M_{age}$ = 15.12 years, $SD$ = 1.49. | Cross-sectional | China | Age, gender, family construct, paternal and maternal education, problem-solving, rumination, depressive symptoms, relationship factors, communication problems, and perceived burdensomeness. | Single item measures. SI: "Have you had suicidal ideation in the past 12 months?". SA: "Have you attempted suicide in the past 12 months?" | Male gender, adolescents from incomplete families, adolescents whose parents had lower educational levels, less rumination, more severe depressive symptoms, less communication problems with parents. | The SA group were significantly less likely to be female (OR = 0.52, 95% CI 0.46–0.60), and more likely to be from incomplete families (OR = 1.40, 95% CI 1.14–1.61), have parents with lower educational levels (i.e., only completing primary schools or without any diploma; maternal: OR = 1.18, 95% CI 1.01–1.39; paternal: OR = 1.41, 95% CI 1.19–1.68), ruminate less (OR = 0.96, 95% CI 0.95–0.97), more severe depressive symptoms (OR = 1.18, 95% CI 1.15–1.20), and less communication problems with their parents (OR = 0.99, 95% CI 0.98–1.00) compared to the SI group. |

*(Continued)*

| Author and date | Sample type and demographics | Study design | Country | Exposure: risk or protective factors measured | Suicide measure | Significant risk or protective factors for suicide attempts | Study findings |
|---|---|---|---|---|---|---|---|
| Yang et al. [145]. | 1596 undergraduates (20.68% female), aged 18–35 years ($M_{age}$ = 20.98 years, $SD$ = 2.88). SI group: $N$ = 278 (26.26% female). SA group: $N$ = 29 (37.9% female). No SI or SA group: $N$ = 1289 (19.08% female). | Cross-sectional | China | Emotion reactivity, emotion dysregulation, depression, and anxiety. | SI: BSS-Chinese version (measures current suicidal ideation) [120,146]. SA: SITBI [45]. | Emotion dysregulation subscales: impulse, control difficulties, emotional awareness, and emotional clarity. | Impulse control difficulties (OR = 1.18, 95% CI 1.04–1.35, $p$ = .01), emotional awareness (OR = 1.15, 95% CI 1.03, 1.28, $p$ = .01), and emotional clarity (OR = 0.78, 95% CI 0.66–0.92, $p$ = .00) significantly distinguished the SA group from the SI group. |
| You & Lin [147]. | Students in grades 7–10 aged 12–18 years ($M_{age}$ = 14.63 years, $SD$ = 1.25) at baseline. 3600 students were followed over three waves. | Longitudinal, baseline (Wave 1), 6-months (Wave 2), and 12 months (Wave 3). | China | NSSI | Single item measures. SI: "Have you had suicidal ideation in the past 12 (Wave 1) or 6 months (Wave 2 and Wave 3)?" SA: "Have you attempted suicide in the past 12 (Wave 1) or 6 months (Wave 2 and Wave 3)?" | NSSI (total sample and females only). | For the total sample, the interaction between NSSI frequency and significantly predicted greater likelihood of suicide attempts ($b$=0.08, OR = 1.08, 95% CI 0.90–0.96). For females only, the interaction between NSSI and SI also predicted suicide attempts ($b$=0.09, OR = 1.10, 95% CI 1.07–1.13). However, the interaction of NSSI and SI did not significantly predict SA in males ($b$=0.00, OR = 1.00, 95% CI 0.91–1.09). |

*(Continued)*

| Author and date | Sample type and demographics | Study design | Country | Exposure: risk or protective factors measured | Suicide measure | Significant risk or protective factors for suicide attempts | Study findings |
|---|---|---|---|---|---|---|---|
| Zhong et al. [148]. | Two samples: Young adults recruited from the Collaborative Psychiatric Epidemiology Survey (CPES), aged 18–30 years. SA group: $N$=303 (61.7% female), $M_{age}$ =23.5 years, $SD$=0.3, 63.4% Caucasian. SI group: $N$=451 (53% female), $M_{age}$ =23.1 years, $SD$=0.4, 70.4% Caucasian. Control group: $N$=3671 (50% female), $M_{age}$ =23.8 years, $SD$=0.1, 61.1% Caucasian. Psychiatric inpatients aged 15–30 years. SA group: $N$=38 (54.1% female), $M_{age}$ =22.8 years, $SD$=3.8, 90% Caucasian. SI group: $N$=40 (44.7% female), $M_{age}$ =22.1 years, $SD$=3.9, 75% Caucasian.. Healthy controls: $N$=37 (27.5% female), $M_{age}$ =22.1 years, $SD$=2.2, 84% Caucasian. | Cross-sectional | US | Age, gender, race, SES, depression, anxiety, PTSD current and lifetime smoking, alcohol drinking, body mass index, childhood abuse, blood pressure, cardiovascular risk, cardiovascular disease, CPES risk scores. | CPES sample: WMH-CIDI [89]. Examined history of SA and SI. Inpatient sample: C-SSRS [53]. | Increased cardiovascular risk, more likely to report established cardiovascular disease, lower SES (inpatient sample), CPES risk scores, high BP, current smokers and lifetime smoker. | The SA group reported lower SES than the SI group (inpatient sample only). In the CPES sample, the SA group reported significantly larger cardiovascular risk scores than the SI group in post hoc comparisons. In the inpatient sample, only SA showed increased cardiovascular risk ($\beta$=1.61, 95% CI 0.53–2.68, Cohen's $d$=0.28, $p$=0.004). The SA group were more likely to be classified in the high cardiovascular risk group (CPES sample: OR = 3.36, 95% CI 1.67–6.78, $p$=0.001; inpatient sample: OR = 9.89, 95% CI 1.38–85.39, $p$=0.03) compared to the SI group (CPES sample: OR = 1.15, 95% CI 0.55–2.39, $p$=0.71; inpatient sample: OR = 1.91, 95% CI 0.25–15.00, $p$=0.53). Blood pressure and current smoking status were significantly higher in the SA group compared to the SI group in both the CPES and inpatient sample. Controlling for covariates, the SA group were more likely to have high blood pressure and to be lifetime smokers. Only the SA group were significantly more likely to be classified in the high-risk group (CPES sample: OR = 3.36, 95% CI 1.67–6.78, $p$=0.001; inpatient sample: OR = 9.89, 95% CI 1.38–85.39, $p$=0.03) |

Note. Study findings are based on multivariate analyses where available. OR = odd ratio; aOR = adjusted odd ratio; aRR=adjusted relative risk; HR=Hazard ratio; CI=confidence interval; ADHD=Attention-deficit hyperactivity disorder; ASD=Autism Spectrum Disorder; ASIQ=Adult Suicide Ideation Questionnaire; BD=Bipolar Disorder; BD-NOS=Bipolar Disorder – Not Otherwise Specified; BSS=The Beck Scale for Suicidal Ideation; C-CASA=Columbia Classification Algorithm for Suicide Assessment; C-SSRS=Columbia-Suicide Severity Rating Scale; CIDI=Composite International Diagnostic Interview; C-DISC=Computerized Diagnostic Interview Schedule for Children; CPES=Collaborative Psychiatric Epidemiology Survey; dACC=dorsal anterior cingulate; Depressive Symptom Inventory-Suicidality Subscale (DSI-SS); DISC=Diagnostic Interview Schedule for Children; dlPFC=dorsolateral prefrontal cortex; dmPFC=dorsomedial prefrontal cortex; GAD=generalised anxiety disorder; HPA axis=Hypothalamic-Pituitary-Adrenal axis; K-SADS-PL=Kiddie Schedule for Affective Disorders and Schizophrenia – Present and Lifetime version; MDD=Major Depressive Disorder; M.I.N.I=Mini International Neuropsychiatric Interview; MRI=Magnetic Resonance Imaging; NSSI=Non-suicidal self-injury; OCD=Obsessive Compulsive Disorder; ODD=Oppositional Defiant Disorder; PHQ=Patient Health Questionnaire; rACC=rostral anterior cingulate cortex; RRR=relative risk ratio; SA=suicide attempt; SBQ-R=Suicide Behaviour Questionnaire-Revised; SITBI=The Self-Injurious Thoughts and Behaviors Interview; SES=socioeconomic status; SIQ=Suicide Ideation Questionnaire; SIQ-JR=Suicide Ideation Questionnaire-Junior; SSRI=Selective-Serotonin Reuptake Inhibitors; SUD=substance use disorder; WHO=World Health Organization; WMH-CIDI=World Mental Health Composite International Diagnostic Interview.

Table 3. Study summary table for self-harm ideation and behaviour.

| Author and date | Sample type and demographics | Study design | Country | Exposure: risk or protective factors measured | Self-harm measure | Significant risk or protective factors for suicide attempts | Study findings |
|---|---|---|---|---|---|---|---|
| Del Carpio et al [149]. | Time 1: 185 general population adolescents/ pupils (52.43% female, 45.95% male, 1.62% undisclosed), aged 11–17 years, $M_{age}$ = 13.16 years, $SD$ = 1.49. Time 2: 115 adolescents (58.26% female, 40% male, 1.74% undisclosed), age 12–18 years, $M_{age}$ = 13.65 years, $SD$ = 1.52. Self-harm ideation: $N$ = 44 (Time 1), $N$ = 26, (Time 2). Self-harm: $N$ = 38 (Time 1), $N$ = 33 (Time 2). | Longitudinal (6-months) | Scotland, UK | Defeat, entrapment, adaptive, coping, maladaptive coping, self-esteem, social support from family, friends, and significant others, stigma (stigmatising beliefs about suicide, isolation/ depression, and glorify-ing/ normalising beliefs about suicide), suicide death, non-suicide death, family self-harm, and friend self-harm. | Self-harm ideation/ SI: SITBI [45] "Have you ever had thoughts of purposely hurting yourself without wanting to die? (for example, cutting or burn-ing)" and "Have you ever had thoughts of killing yourself?" respectively. Self-harm: Items from a CASE study question-naire: "Have you ever deliberately taken an overdose (e.g., of pills or other medication) or tried to harm yourself in some other way (such as cut yourself)?" [150]. | Glorifying/ normal-ising beliefs about suicide (baseline only), family self-harm (baseline only). | Baseline (Time 1): The NSSI group were signifi-cantly less likely to endorse beliefs about suicide (aOR = 0.42, 95% CI 0.22–0.80), than the NSSI ideation group. In the univariate analyses, the NSSI group was significantly more likely to report family self-harm (OR = 0.17, 95% CI 0.04–0.70; no multivariate analyses conducted). At follow-up the NSSI group did not differ from the follow-up ideation group on any variables. |
| Garcia-Neito et al [151]. | 267 youth (34.7% female), aged 11–18 years, $M_{age}$ = 14.1 years, $SD$ = 1.9. NSSI group: $N$ = 58. NSSI ideation group: $N$ = 39. No NSSI ideation or behaviour group: $N$ = 142. | Retrospective, cross-sectional | Spain | Strengths and difficulties (emotional problems, behavioural problems, hyperactivity, peer problems and prosocial behaviour), trait/state anger (anger/ trait total, externalization of anger, internalization of anger, external control, internal control), depression (total, dysphoria, and self-esteem), global assessment and global impression (i.e., severity of a patients' sympto-mology, completed by clinicians). | Spanish version of the SITBI [45,152]. | Internalization of anger, depression, worse overall functioning. | The NSSI group had signifi-cantly greater internaliza-tion of anger, compared to all other groups ($F$ = 4.1 (2), $p$ = .018). The NSSI group also scored significantly higher on the Children's Depression Inventory ($F$ = 11.32 (2), $p$ < .001). The NSSI group also displayed worse overall functioning compared to the NSSI ideation group. |

*(Continued)*

**Table 3.** (Continued)

| Author and date | Sample type and demographics | Study design | Country | Exposure: risk or protective factors measured | Self-harm measure | Significant risk or protective factors for suicide attempts | Study findings |
|---|---|---|---|---|---|---|---|
| Jiang et al. [153]. | 606 students (38.8% female) aged 11–16 years ($M_{age}$ = 13.58 years, $SD$ = 1.04),. NSSI ideation group: $N$ = 98 (39.8% female). NSSI group: $N$ = 86 (48.8% female). No NSSI group: $N$ = 422 (36.49% female). | Cross-sectional | China | Six subscales of the self-compassion scale (i.e., self-kindness/ self-judgement, common humanity/ isolation, and mindfulness/ over identification). | NSSI ideation: "Have you ever thought about injuring yourself, like cutting, burning, and banging, without the intent to die, in the past year?". NSSI: "Have you ever engaged in self-injury, like cutting, burning, and banging, without the intent to die, in the past year?" | Less self-kindness and common humanity (i.e., recognising one's distress as part of the human experience) subscales of a self-compassion scale. | The NSSI group demonstrated less self-kindness ($p$ = 0.001) and common humanity ($p$ = 0.006) than the NSSI ideation group. |
| McMahon et al [154]. | 3684 adolescent students (52% female), aged 14–17 years. Three groups: History of self-harm group, self-harm thoughts group, and no self-harm thoughts or acts group. | Cross-sectional | Southern Ireland | Emotion-oriented coping and problem-oriented coping. | Self-harm: At least one of the following acts: (1) Initiated behaviour (e.g., self-cutting) with the intent to cause harm to self, (2) ingested a substance in excess, (3) ingested a recreational or illicit drug which was regarded as self-harm, (4) ingested a non-ingestible substance. Self-harm thoughts: During the past month or the past year, adolescents seriously thought about taking an overdose or trying to harm themselves without actually doing so. Did not provide a clear distinction between suicidal and NSSI [155]. | Greater emotion-oriented coping, lower problem-oriented coping (females only). | Female adolescents with NSSI reported significantly greater emotion-oriented coping ($p$ = 0.001) and significantly lower problem-oriented coping ($p$ = 0.014) compared to the NSSI ideation group. |

*(Continued)*

**Table 3.** (Continued)

| Author and date | Sample type and demographics | Study design | Country | Exposure: risk or protective factors measured | Self-harm measure | Significant risk or protective factors for suicide attempts | Study findings |
|---|---|---|---|---|---|---|---|
| Madge et al [156]. | 30,477 adolescents aged 14–17-years, majority of sample 15–16 years (47.8% female). Four groups: Self-harm thoughts only, single self-harm episode (one episode in the past year), multiple self-harm episode (an episode in the past year and at least one other episode), and no self-harm thoughts or behaviour group. | Cross-sectional | Seven countries: Belgium, England, Hungary, Ireland, Netherlands, Norway and Australia. | Age, gender, country, anxiety, depressive symptoms, self-esteem, impulsivity, and stressful life events (consisting of ten categories including (1) difficulties with friends and peers, (2) problems with or between parents, (3) serious illness of family or friend, (4) physical or sexual abuse, (5) suicide or self-harm of family or friend, (6) death of someone close, (7) worries about sexual orientation, (8) trouble with police, (9) being bullied, and (10) problems with schoolwork). | Self-harm: At least one of the following acts: (1) Initiated behaviour (e.g., self-cutting) with the intent to cause harm to self, (2) ingested a substance in excess, (3) ingested a recreational or illicit drug which was regarded as self-harm, (4) ingested a non-ingestible substance. Self-harm thoughts: During the past month or the past year, adolescents seriously thought about taking an overdose or trying to harm themselves without actually doing so. Did not provide a clear distinction between suicidal and NSSI [155]. | Female gender, higher impulsivity, experiencing the suicide or self-harm of others, physical or sexual abuse, worries about sexual orientation. | Being female, (p=0.001), impulsivity (p=0.005) and experience of the suicide/self-harm of others (p<0.001) significantly distinguished adolescents with a single self-harm episode from those with self-harm ideation only. Physical or sexual abuse (p<0.001) and worry regarding sexual orientation (p=0.017) also significantly distinguished between adolescents with a single self-harm episode from those with self-harm ideation only. |
| O'Connor, Rasmussen, & Hawton [24]. | 5604 population adolescents (49.53% female, 50.34% male, 0.12% undisclosed). 90% were 15–16 years. SI group (N=675). SA group (N=628). | Cross-sectional | Scotland and Northern Ireland, UK | Socially prescribed perfectionism, brooding, self-esteem, optimism, having a family member who had engaged in self-harm, having friends who have engaged in self-harm, descriptive norms, impulsivity, negative life stress. | NSSI ideation: "Have you ever seriously thought about taking an overdose or trying to harm yourself but not actually done so?" NSSI: "Have you ever seriously thought about taking an overdose or trying to harm yourself but not actually done so?" | Female gender, have a family member or close friend who have self-harmed, more likely to think their peers engaged in self-harm, greater impulsivity, greater life stress. | The NSSI group were significantly more likely to be female (aOR = 1.51, 95% CI 1.10–2.08), have a family member (aOR = 2.60, 95% CI 1.79–3.77) or friend (aOR = 1.80, 95% CI 1.28–2.52) who had engaged in self-harm, and to have experienced negative life stress (aOR = 1.05, 95% CI 1.01–1.08), than the NSSI ideation group. In addition, depression approached significance when distinguishing the NSSI group form the NSSI ideation group (aOR = 1.04, 95% CI 1.00–1.08). |

*(Continued)*

**Table 3.** (Continued)

| Author and date | Sample type and demographics | Study design | Country | Exposure: risk or protective factors measured | Self-harm measure | Significant risk or protective factors for suicide attempts | Study findings |
|---|---|---|---|---|---|---|---|
| Saarijarvi et al. [157]. | 118 adolescent patients, aged 12–22 years (Median age = 15 years at Time 1 and 16 years at Time 2). | Longitudinal | Finland | Early maladaptive schemas of emotional deprivation, abandonment, mistrust/abuse, social isolation, defectiveness, failure, dependence, vulnerability, enmeshment, subjugation, self-sacrifice, emotional inhibition, unrelenting, standards, entitlement, insufficient self-control, approval-seeking, negativity/ pessimism, punitiveness. | Ottawa Self-Injury Inventory [158]. | Early maladaptive schemas of emotional deprivation, mistrust/abuse, social isolation, defectiveness, failure, dependence and insufficient self-control. | The self-harm thoughts and behaviour group had significantly greater early maladaptive schemas of emotional deprivation ($p = .04$, −0.28), mistrust/abuse ($p = .04$, −0.26), social isolation ($p = .01$, −0.36), defectiveness ($p = .02$, −0.32), failure ($p = 0.00$, −0.41), dependence ($p = .01$, −0.33) and insufficient self-control ($p = 0.03$, −0.30). |

Note. Study findings are based on multivariate analyses where available. OR = odd ratio; aOR = adjusted odd ratio; HR = Hazard ratio; CI = confidence interval; NSSI = Non-suicidal self-injury; SI = suicidal ideation; SITBI = The Self-Injurious Thoughts and Behaviors Interview. MacMahon et al. [154] presents data from Ireland as part of the multi-centre Child & Adolescent Self-harm in Europe (CASE) study [155,156].

## Substance use

Thirty studies examined substance use as a risk factor. Eighteen studies found greater substance use (e.g., cocaine use) to significantly distinguish between the suicide attempts and suicidal ideation groups [59,84,136], However, when individual substances were examined in some studies, findings were significant for specific substances only (e.g., marijuana use [59]). Of the eighteen studies, one study examined substance use as one of seven indicators of suicide capability and found that the relationship between suicidal ideation and suicide attempts varied as a function of suicide capability [136]. Conversely, 11 studies found that substance use did not distinguish suicide attempts from suicidal ideation (e.g. [121],). One additional cross-sectional study of American Indian youth found those with a greater proportion of their social network that use alcohol or drugs or that they use alcohol or drugs with significantly distinguished suicide attempts from suicidal ideation [77].

## Alcohol use

Twenty studies examined alcohol use. Five studies found greater alcohol use and alcohol-related disorders to significantly distinguish suicide attempts from suicidal ideation [31,35,59,81,142]. The remaining fifteen studies found that alcohol use did not significantly distinguish between suicide attempts and suicidal ideation (e.g. [84,95,127],). One of these studies, however, found that the interaction between major depressive disorder (MDD) severity and frequency of alcohol use significantly distinguished suicide attempts from suicidal ideation [47].

## Tobacco use/ smoking

Fourteen cross-sectional studies examined smoking. Eleven studies found greater cigarette smoking to significantly distinguish suicide attempts from suicidal ideation (e.g. [61,83,140],). Whereas, three studies found no significant differences [80,81,84].

## Emotional, physical or sexual abuse

Twelve studies examined abuse as a risk factor. Ten studies found at least one form of emotional, physical or sexual abuse (including childhood, recent, or lifetime abuse) to significantly distinguish suicide attempts from suicidal ideation (e.g. [44,92,135],). However, the type of abuse that significantly distinguished between suicide attempts and suicidal ideation were inconsistent across studies. In contrast, two studies found no forms of abuse to significantly distinguish suicide attempt and ideation groups [11,12].

## Suicidal ideation frequency, severity and suicidal plans

Fourteen studies examined suicidal ideation, frequency and severity and suicidal plans. Nine cross-sectional studies found greater suicidal ideation severity and duration and suicide plans to significantly distinguish suicide attempts from suicidal ideation (e.g. [40,42,96],). Conversely, four cross-sectional and one longitudinal study did not find greater suicidal ideation/ presence of suicide plans to distinguish suicide attempts from suicidal ideation (e.g. [11,113,132],).

## Exposure to suicide or self-harm in friends, family, or significant others

Nine studies examined exposure to suicide or self-harm in others. Four cross-sectional studies and one longitudinal study identified that exposure to suicide or self-harm in others significantly distinguished suicide attempts from suicidal ideation [11,44,95,134,143]. However, three cross-sectional studies [77,105,122] and one longitudinal study [103] found no significant differences.

## Violence

Nine cross-sectional studies of adolescents examined violence as a risk factor. Of these, seven found that various forms of violence (e.g., weapon violence victimization; being physically attacked) significantly distinguished suicide attempts

from suicidal ideation [47,59,85,97,119,127,133]. However, notably not all forms of violence were significant across studies (e.g., dating violence [97,127,133]). Vélez-Grau et al. [136] found that no measures of violence significantly distinguished suicide attempts from suicidal ideation. One additional study found that receiving treatment for violent experiences significantly distinguished the suicide attempt group from the suicidal ideation group [31].

## Impulsivity

Eleven studies examined impulsivity. Four cross-sectional studies found that higher levels of state or trait impulsivity significantly distinguished suicide attempts from suicidal ideation [66,80,95,143]. However, six studies (five cross-sectional and one longitudinal) did not find impulsivity to distinguish suicide attempts from suicidal ideation [11,31,68,96,98,133]. Valderrama et al. [135] in a cross-sectional study of undergraduates, found that only lack of perseverance distinguished suicide attempts from suicidal ideation, whereas no other impulsivity dimensions distinguished between groups.

## Depression

Twelve cross-sectional studies found that depressive disorders/symptoms significantly distinguished suicide attempts from suicidal ideation (e.g. [31,65,119],). Of these, studies suggest the severity of depression is important (e.g. [42],) and identified more severe diagnoses, e.g., MDD/ major depressive episodes as risk factors (e.g. [80],). However, twenty-two studies did not find that depressive disorders/symptoms significantly distinguished suicide attempts from suicidal ideation (e.g. [103,104,121],). One study found that the interaction between frequency of alcohol use and MDD significantly distinguished suicide attempts from suicidal ideation [47]. Conversely, Stewart et al. [132] found less severe depressive symptoms to distinguish suicide attempts from suicidal ideation.

## Family functioning

Across nine studies, family factors (e.g., lower parental care and worsening recent relationship quality with parents) significantly distinguished suicide attempts from suicidal ideation (e.g. [83,125,126],). However, one study found that lower parental care predicted suicide attempts in a psychiatric inpatient sample, but not in a high school sample [120]. Four cross-sectional studies found that family factors (e.g., living with a single parent), did not significantly distinguish suicide attempts from suicidal ideation (e.g., family living together, family involvement [69,81,84,125];).

## Stressful life events

Four cross-sectional studies examined stressful life events. Two studies of adolescents found that various stressful life events (e.g., serious illness, accident, abuse) significantly distinguished suicide attempts from suicidal ideation [105,106]. Stewart et al. [132] found only interpersonal loss life events to distinguish suicide attempts from suicidal ideation in adolescent inpatients. By contrast, Macrynikola et al. [93] found that stressful life events did not significantly distinguish suicide attempts from suicidal ideation in college students.

## Bullying experiences

Nine studies examined bullying experiences. Of these, four studies found that bullying experiences (i.e., being a perpetrator or victim, being kicked, pushed or shoved or made fun of about sex, and being cyberbullied) significantly distinguished suicide attempts from suicidal ideation [59,84,117,137]. However, other types of bullying experiences (e.g., being left out of activities, verbal bullying) examined in these studies did not significantly distinguish suicide attempts from suicidal ideation. By contrast, five cross-sectional studies found no significant differences (e.g. [31,96,97],).

### Emotion dysregulation

Four cross-sectional studies [68,80,120,145] examined emotion dysregulation. Of these, two studies found that emotion dysregulation significantly distinguished suicide attempts from suicidal ideation [80,145], whereas two studies found no such difference [68,120].

### Physical health

Two cross-sectional studies examined physical health problems and found that cardiovascular disease distinguished suicide attempts from suicidal ideation [60,148].

### Brain structure and function

One study examined markers of brain structure [75] and four studies examined markers of brain function [42,104,109,128] using MRI. Of these studies various markers of neural functioning (e.g., amygdala connectivity, amygdala functioning, lower bilateral caudate activity during positive self-processing) and structure (e.g., right lateral orbitofrontal thickness, left fusiform thickness, and left temporal pole volume) significantly distinguished suicide attempts from suicidal ideation, whereas others did not (e.g., caudate activity during negative vs. positive self-appraisals; left caudal anterior cingulate thickness and right rostral anterior cingulate volume).

### Biological markers

Five studies examined additional biological markers [38,92–95]. Various biological markers, including greater monocyte count, blunted cortisol reactivity, and lower eosinophil percentages, significantly distinguished suicide attempts from suicidal ideation. However, several other biological markers (e.g., GR-β mRNA, GR sensitivity, and 1L-1β mRNA) did not (e.g. [99],).

## Protective factors distinguishing suicide attempts from suicidal ideation

### Academic achievement/ Intelligence

Three cross-sectional studies examined academic achievement in adolescent students and found that greater academic achievement significantly distinguished suicidal ideation from suicide attempts [33,133,140]. Likewise, Kwon et al. [31] found low academic performance to significantly distinguish suicide attempts from suicidal ideation. One study examining IQ found no significant differences [122].

### Parental factors

One cross-sectional study of adolescent students found that parental factors (e.g., parental connectedness) significantly distinguished suicidal ideation from suicide attempts [105].

### Help-seeking

One longitudinal study of adolescents found that for those reporting suicidal ideation, help-seeking behaviours at baseline reduced the onset of suicide attempts over time [62]. However, this finding emerged in Black males and Latinas only.

### Anxiety sensitivity

A cross-sectional study of adolescents found that anxiety sensitivity reduced the likelihood of making a suicide attempt among adolescents reporting suicidal ideation [98].

**Summary of factors distinguishing suicide attempts from suicidal ideation**

Overall, findings were mixed across studies. The most consistent risk factors that distinguished suicide attempts from suicidal ideation included being female, NSSI, physical, emotional, or sexual abuse, violence, and family factors. Although inconsistent, other risk factors identified included age, emotion dysregulation, stressful life events, impulsivity, substance use, bullying, biological markers (e.g., blunted cortisol reactivity), physical health problems (e.g., cardiovascular disease), preferential information processing, affective instability, mental health disorders, psychological distress, suicidal ideation frequency, severity and planning, and unhealthy coping strategies, among others. Protective factors identified included parental connectedness, academic achievement, neighbourhood safety, White ethnicity (in males only) and living with both biological parents (in females only).

**Study characteristics of studies examining risk and protective factors distinguishing self-harm behaviours from self-harm ideation**

Seven studies distinguished self-harm behaviour from self-harm ideation. Sample sizes ranged from 185 to 30,477. Studies were conducted in the UK ($N=2$; Scotland and Northern Ireland), China ($N=1$), Southern Ireland ($N=1$), and Spain ($N=1$). One study was conducted across seven countries (Belgium, England, Hungary, Ireland, Netherlands, Norway and Australia). All studies were quantitative; five studies were cross-sectional, and two were longitudinal [149,157]. No qualitative studies met the inclusion criteria. Three studies employed single-item measures to assess self-harm ideation and behaviour, two did not report a specific measure, and two utilised a validated multi-item measure of self-harm.

**Methodological quality**

Quality assessment scores for studies examining self-harm ideation and behaviour are reported in the supplementary material (see S3 Table). The maximum score obtainable was 12. The mean score across studies examining self-harm ideation and behaviours was $5.43\pm2.15$ (range 2–9). The lowest scoring domain was power calculation, followed by study design.

**Risk factors distinguishing self-harm-behaviours from self-harm ideation**

**Gender**

Two cross-sectional studies of adolescents found that being female significantly distinguished self-harm behaviours from self-harm ideation [24,156].

**Exposure to self-harm/ suicide**

Three studies in adolescents found that exposure to suicide or self-harm of others distinguished self-harm behaviours from self-harm ideation. Madge et al. [156] examined exposure to suicide/self-harm of others, whereas two other studies examined exposure to self-harm in family or friends [24,149]. O'Connor et al. [24] found that exposure to self-harm in a family member or close friend significantly distinguished self-harm behaviours from self-harm ideation. Whereas, Del Carpio et al. [149] found that only exposure to self-harm in family significantly distinguished self-harm behaviours from self-harm ideation in univariate analyses (no multivariate analyses was conducted). However, this finding did not replicate when examined prospectively.

**Impulsivity**

Two cross-sectional studies examining trait impulsivity in adolescents found greater impulsivity to significantly distinguish self-harm behaviours from self-harm ideation [24,156]. No studies examined state impulsivity.

**Early maladaptive schemas**

One study examined early maladaptive schemas and found that the self-harm behaviour group had significantly greater early maladaptive schemas of emotional deprivation, mistrust/abuse, social isolation, defectiveness, failure, dependence and insufficient self-control, compared to the self-harm ideation group [157].

**Summary of factors that distinguish self-harm behaviours from self-harm ideation**

While research was lacking on risk factors that distinguished self-harm behaviours from self-harm ideation, the most identified risk factors included being female, exposure to self-harm/ suicide, and impulsivity. Additional risk factors included internalization of anger, worse overall functioning, lower self-compassion, lower problem-oriented coping and greater emotion-oriented coping (in females only), worries about sexual orientation and greater life stress. No included studies on self-harm examined protective factors.

## Discussion

The aim of this review was to synthesize findings from existing research studies which have examined risk or protective factors in young people that distinguish between those who have thoughts of self-harm or suicide from those who act on their thoughts. Findings were inconsistent across studies. Risk factors that distinguished suicide attempts from suicidal ideation included being female, NSSI, emotional, physical and sexual abuse, violence, and family factors (e.g., conflict). There was also some evidence which found substance/alcohol use, suicidal severity, frequency and planning, emotion dysregulation, and life stress to distinguish suicide attempts from suicidal ideation. Protective factors, including parental factors (e.g., parental connectedness) and academic achievement, distinguished suicidal ideation from suicide attempts. In contrast to the vast number of studies examining factors that distinguish suicide attempts from suicidal ideation, only seven studies examined factors that distinguish self-harm behaviour from self-harm ideation. Here, being female, exposure to self-harm/suicide, and impulsivity significantly distinguished self-harm behaviour from self-harm ideation; however no protective factors were examined.

Notably, the presence of NSSI emerged as the most consistent risk factor that distinguished suicide attempts from suicidal ideation. This finding aligns with a meta-analytical review which found self-harm ideation and behaviours to significantly predict suicide attempts and deaths over time and with research suggesting that self-harm is one of the strongest predictors of future suicide [159,160]. Here, NSSI may represent a clear indication of distress, which if not dealt with, can increase over time. The repeated act of inflicting pain and physical damage on oneself may also desensitise individuals to pain and increase fearlessness of death, particularly because NSSI in young people often escalates in severity over time (e.g. [161],). Consistent with this, while only examined in two studies, Bayliss et al. [25] found interoceptive deficits to distinguish suicide attempts from suicidal ideation. This decreased ability to perceive bodily sensations may increase one's capacity to engage in NSSI and more lethal self-injury, and subsequently increase the likelihood of transitioning from suicidal thoughts to behaviours (see [16]). Given the prevalence of NSSI in young people (e.g. [4],) and evidence from the current review, NSSI remains an important target for preventative efforts to prevent escalation to suicidal behaviours.

While inconclusive, substance abuse, in addition to physical, emotional, and sexual abuse, were often identified as significant risk factors in distinguishing suicide attempts from suicidal ideation in young people. This finding aligns with May and Klonsky's meta-analytic review [10] where both drug use disorder and sexual abuse distinguished suicide attempts from suicidal ideation in adults, albeit with modest effect sizes. Likewise, this finding also aligns with Bayliss et al.'s systematic review [25] that found painful and provocative events (e.g., abuse, maltreatment) to significantly distinguish suicide attempts from suicidal ideation in adults. Both substance abuse and physical, emotional, and sexual abuse are associated with increased provocative and painful events which may increase the likelihood of moving from suicidal thoughts to a suicide attempt [17]. Substance abuse may also alter logical thinking and increase the likelihood of engaging

in impulsive behaviours. In addition, May and Klonsky [10] suggest that drug use disorders may play a role in the transition due to greater painful physical symptoms (e.g., withdrawal), involvement in risky situations and tolerance to self-inflicted pain (e.g., injecting drugs, overdose), increasing capability of suicide. A departure to May and Klonsky's review [10], however, is that in the current review, physical and emotional abuse appeared to be a risk factor for suicide attempts in young people, whereas alcohol use did not. These findings indicate possible differences in risk factors between adults and young people.

The severity and frequency of suicidal ideation and the presence of a suicidal plan to distinguish suicide attempts from suicidal ideation emerged as inconsistent across studies in the review (e.g. [11,42],). While suicidal ideation and suicidal plans are well-known predictors of suicide, only suicidal planning has been proposed as a risk factor involved in the transition from suicidal ideation to attempts in some theoretical models [19]. In contrast, all theoretical models of suicide suggest that suicidal ideation progresses to suicide attempts only when one has the capacity to attempt suicide [17,19,20]. Here, it is likely the case that the severity or frequency of suicidal ideation or the presence of a suicide plan alone is not sufficient to transition from thoughts to attempts. It is, therefore, important to consider other contributors, particularly those in regard to capability for suicide, alongside the severity and frequency of suicidal ideation, and suicidal planning.

The ideation-to-action framework is important given that many previously identified risk factors for suicide (e.g., depressive disorders/symptoms) are considered risk factors for suicidal ideation, but not suicide attempts and do not distinguish between these two groups (see [9]). In the current review, findings with depressive disorders/symptoms were mixed. This finding is not surprising based on previous meta-analyses (e.g. [10,162],). Ribeiro et al.'s [162] review, for instance, found that depression conferred risk for suicide ideation, suicide attempts, and suicide death, but found greatest support for depression as a predictor of suicidal ideation. In line with this, May and Klonsky's meta-analytic review [10] suggests that depression does play a small role in distinguishing suicide attempts from suicidal ideation, but likely plays a bigger role in distinguishing suicidal ideation from control groups given that differences in the prevalence of mental disorders are often much greater between suicidal ideation and control groups, than between suicidal ideation and attempt groups. Depressive disorders/symptoms may, therefore, be somewhat important in facilitating the transition from suicidal ideation to attempts, but alone may not be a key risk factor, and should be considered in combination with other factors.

In this review, findings of state and trait impulsivity were mixed. In studies that found impulsivity to play a role, both state and trait impulsivity distinguished suicide attempts from suicidal ideation (e.g. [95],). While impulsivity is an oft-cited risk factor involved in the transition from suicidal ideation to attempts (e.g. [19],), its role is not consistently supported (see [14]). This inconsistency in findings may be partly explained by the variation in its conceptualization and measurement (see [163]). Impulsivity, for instance, is a multidimensional construct which can refer to both trait and state components and distinct dimensions (e.g., cognitive, behavioural and mood-based impulsivity; see [163]). More attention should be placed on the distinct dimensions of impulsivity given that dimensions vary in their relation to suicidal behaviour (e.g. [164],).

There was, however, no clear pattern to explain why some studies found impulsivity to be important in this transition and other studies did not. Regarding state impulsivity, the timing of measurement seems to be particularly important (i.e., whether impulsivity was measured close to an attempt). For example, Liu et al. [165] found that the length of time between measurement of impulsivity and suicide attempts moderated the relationship between impulsivity and suicide attempts, where the closer the timing of measurement, the stronger the effect. Similarly, using sequence analysis, Townsend et al. [166] have shown that impulsivity is the only factor (from a wide range of risk factors) that is directly proximal to an act of self-harm in young people. Future research is required to disentangle the nature of various dimensions of impulsivity in the transition from suicidal ideation to attempts in young people. Ecological momentary assessment studies may be particularly useful in examining changes in state impulsivity prior to a suicide attempt.

This review offered some indication that exposure to self-harm or suicidal behaviours in family or friends may be a risk factor distinguishing suicide attempts from suicidal ideation, however findings were inconsistent [44,105,143]. Exposure

to self-harm and suicidal behaviour has previously been identified as an important risk factor involved in the transition between suicidal ideation and attempts (see [19]). While one study included in this review found that exposure to suicide in friends (but not family) was significantly associated with suicide attempts [143], no consistent patterns emerged across studies in the review. Prior research suggests that exposure to self-harm or suicide in friends may play a more important role in young people who tend to be greatly influenced by their peers and are prone to social contagion. Future research contrasting adolescent and adult samples' exposure of suicide and self-harm in friends, family, and significant others is warranted. Hawton et al. [167] has noted concerns regarding online exposure to self-harm/ suicide and social contagion in young people. This was evident in Liu et al.'s [86] study included in the review where the suicide attempt group reported significantly greater suicide-related social media use behaviours (i.e., attending to suicide information, commenting on or reposting suicide information or talking about suicide), relative to the suicidal ideation group. Future research should prioritise examining whether online exposure to suicide/ self-harm plays an important role in distinguishing suicide attempts from suicidal ideation in this population.

While interpersonal problems are largely considered to be risk factors for suicidal ideation in theoretical models that incorporate an ideation-to-action framework (e.g., perceived burdensomeness [16];), some interpersonal factors in the current review did distinguish suicide attempts from suicidal ideation. For instance, various interpersonal factors involving family (e.g., family conflict) were found to distinguish suicide attempts from suicide ideation across studies (e.g. [125],). Because adolescence is associated with heightened interpersonal sensitivity, interpersonal problems may increase the likelihood of suicide attempts in this population. Future research comparing adult and adolescent samples on various interpersonal factors within the ideation-to-action framework will be informative. Comparison of other samples (e.g., clinical and non-clinical adolescents) is also warranted given that some interpersonal factors (e.g., lower parental care) only distinguished between suicide attempts and suicidal ideation in a psychiatric sample, but not in a non-clinical sample [120].

In comparison to research examining factors that distinguish suicide attempts from suicide ideation, much less research has focused on factors that distinguish between self-harm behaviours and self-harm ideation. Risk factors for self-harm often overlap with risk factors for suicide (see [2]). This was seen in the current review, where some evidence indicated that being female, exposure to self-harm or suicide, and impulsivity distinguished self-harm behaviours from self-harm ideation and suicide attempts from suicidal ideation. Moreover, the risk factors identified here align with considerable evidence suggesting that self-harm is more prevalent in female adolescents [155], and that factors such as social contagion and impulsivity tend to be more specific to youth (see [26,146]). That said, given the lack of research examining factors that distinguish self-harm behaviours from self-harm ideation, the extent to which these findings would replicate across studies is unclear. Future research examining risk and protective factors that distinguish between self-harm behaviours and self-harm ideation is warranted to provide more conclusive evidence.

Our review found that some protective factors including parental factors (e.g., parental connectedness) and academic achievement distinguished between suicidal ideation and suicide attempts [33,133], whereas we found no evidence for protective factors that distinguished self-harm ideation from self-harm behaviours. Hence, for the most part, findings of the review offer support for a selection of risk factors as important in the transition from suicidal thoughts to suicidal behaviour, rather than protective factors. It is important, however, not to over interpret this finding as relatively few studies focused on protective factors. In this regard, the absence of evidence is not the same as finding that protective factors do not play a role. Future research is needed to determine whether protective factors significantly distinguish between ideation and behaviour groups.

## Limitations of the review and future directions

Despite the novel contribution of this review, there are limitations. For example, relevant studies not written in English will have been excluded, limiting generalizations of the findings. In addition, the search was conducted between 2011–2024, therefore relevant studies published before 2011 would not have been retrieved (e.g. [168],). Most studies examining the

"ideation-to-action" framework, however, were likely published after 2011 and this review synthesizes more contemporary evidence on this framework. In addition, we excluded studies that did not represent a true ideation group (i.e., where all or some of participants reported previous history of self-harm/ suicidal behaviours; e.g. [38,169–171],). It is possible that ideation groups with previous history of self-harm/ suicidal behaviour differ from true ideation groups on risk factors (e.g., see [170]). Future research should investigate factors that distinguish those who have history of self-harm/suicidal behaviour but experience current ideation only and those who report current ideation and behaviours. Furthermore, there was considerable heterogeneity of variables, including measurement of risk and protective factors and outcome variables and the timescale of measures (e.g., present, lifetime). Therefore, it was not deemed appropriate to conduct a meta-analysis.

Our review was also limited by most eligible studies utilising cross-sectional designs. Future research examining the ideation-to-action framework should employ prospective designs. In addition, only a few studies included in the review recruited samples from low and middle-income countries (e.g. [59,65,107,121,168],). Given that three quarters of the world's suicides occur in low and middle-income countries and that the greatest burden of self-harm is felt in low-and-middle income countries [172,173], future research examining risk and protective factors that distinguish self-harm behaviours from self-harm ideation or suicide attempts from suicidal ideation in low and middle-income countries is warranted. Our review was also limited by the lack of protective factors examined and by the lack of literature examining risk/protective factors that distinguish self-harm behaviours from self-harm ideation. Future research on the ideation-to-action framework should examine protective factors more broadly and risk/protective factors that distinguish self-harm behaviour from self-harm ideation. Additionally, most included studies relied on self-report measures of psychosocial variables. Future research should examine the role of neural (e.g., adolescent brain maturation) and biological markers in distinguishing between suicidal and self-harm behaviours and suicidal and self-harm ideation. Finally, given the complexity of self-harm and suicidal behaviours, as well as the inconsistencies found in this review, future research should consider examining various combinations of risk and protective factors involved in the transition from thoughts to behaviours.

## Conclusion

The ideation-to-action framework has gained momentum in the suicide and self-harm research literature and is recognised as an important framework to identify factors that distinguish between those with thoughts of self-harm and suicide from those who act on their thoughts. This review was the first to synthesise literature examining risk and protective factors that distinguish suicide attempts from suicidal ideation and self-harm behaviours from self-harm ideation in young people. While findings were mixed, the review identified key risk factors distinguishing suicidal attempts from suicidal ideation, predominantly NSSI, physical, emotional and sexual abuse, family factors (e.g., conflict), and violence. Although research was lacking on risk factors distinguishing self-harm behaviours, some risk factors included being female, exposure to self-harm/suicide, and impulsivity. Protective factors distinguishing between suicidal ideation and suicide attempts included parental factors (e.g., parental connectedness) and academic achievement; however, there was no evidence for protective factors distinguishing self-harm ideation from self-harm behaviours. Future work should target and incorporate empirically supported factors into intervention and prevention efforts.

## Supporting information

**S1 Table. Quality assessment tool and scoring guide.**
(DOCX)

**S2 Table. Quality assessment for suicidal ideation and suicide behaviours.**
(DOCX)

**S3 Table. Quality Assessment for self-harm ideation and self-harm behaviours.**
(DOCX)

**S4 Table. Preferred reporting items for systematic reviews and meta-analyses (PRISMA 2020 Checklist).**
(DOCX)

## Author contributions

**Conceptualization:** Marianne Etherson, Chris Hollis, Ellen Townsend, Dorothee P. Auer, Rory C. O'Connor.

**Data curation:** Marianne Etherson, Sieun Lee, Krystyna J. Loney.

**Formal analysis:** Marianne Etherson.

**Funding acquisition:** Chris Hollis, Ellen Townsend.

**Investigation:** Marianne Etherson, Sieun Lee, Krystyna J. Loney, Rory C. O'Connor.

**Methodology:** Marianne Etherson, Krystyna J. Loney, Rory C. O'Connor.

**Project administration:** Marianne Etherson.

**Supervision:** Rory C. O'Connor.

**Writing – original draft:** Marianne Etherson, Sieun Lee, Krystyna J. Loney, Isobel P. Steward, Joey Ward, Heather McClelland, Aaron Kandola, Josimar A. De Alcantara Mendes, Chris Hollis, Ellen Townsend, Dorothee P. Auer, Rory C. O'Connor.

**Writing – review & editing:** Marianne Etherson, Sieun Lee, Krystyna J. Loney, Isobel P. Steward, Joey Ward, Heather McClelland, Aaron Kandola, Josimar A. De Alcantara Mendes, Chris Hollis, Ellen Townsend, Dorothee P. Auer, Rory C. O'Connor.

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
