## [Decision Letter · Decision Letter 0]

Dear Dr.  Etherson,

An abstract may need subsections (background, objective, methodology, result, and discussion).Clarify the reason why the term ‘protective factors’ has been included in the title since you didn’t find any single study regarding this, as you mentioned in the abstract. Introduction, Paragraph 1, “Despite this prevalence, little is known regarding the distinct risk and protective factors that distinguish young people who have thoughts of self-harm or suicide from those who act on their thoughts. It is, therefore, important to identify factors involved in this transition, with the intention to inform future preventative and treatment efforts.” I recommend authors shift this statement to the last paragraph of the introduction section. Even I can see a similar concept from the last paragraph of the introduction.Please include some evidence regarding the eligibility criteria based on study types (qualitative, quantitative, mixed..). In addition, for the benefit of the readers, it is good to indicate some reasons for restricting studies based on years of publication.No evidence regarding risk or protective factors from your searching terms.I am not sure your article screening approach is appropriate since only first authors conducted overall screening. I hope you have evidence on how authors can screen and extract articles using Covidence or other software together.I am not sure how authors extracted results of qualitative and /or quantitative studies if they considered both.

We look forward to receiving your revised manuscript.

Kind regards,

Alemayehu Molla Wollie

Academic Editor

PLOS ONE

“The authors acknowledge the support of the UK Research and Innovation (UKRI) Digital Youth Programme award which is part of the AHRC/ESRC/MRC Adolescence, Mental Health and the Developing Mind programme. Grant number: MR/W002450/1”

5. One of the noted authors is a group or consortium “Digital Youth Team”. In addition to naming the author group, please list the individual authors and affiliations within this group in the acknowledgments section of your manuscript. Please also indicate clearly a lead author for this group along with a contact email address.

6. Please include captions for your Supporting Information files at the end of your manuscript, and update any in-text citations to match accordingly. Please see our Supporting Information guidelines for more information: http://journals.plos.org/plosone/s/supporting-information .

7. As required by our policy on Data Availability, please ensure your manuscript or supplementary information includes the following:

Reviewers' comments:

Reviewer's Responses to Questions

**Comments to the Author**

1. Is the manuscript technically sound, and do the data support the conclusions?

Reviewer #1: Yes

Reviewer #2: Yes

2. Has the statistical analysis been performed appropriately and rigorously?

Reviewer #1: Yes

Reviewer #2: Yes

3. Have the authors made all data underlying the findings in their manuscript fully available?

Reviewer #1: Yes

Reviewer #2: Yes

4. Is the manuscript presented in an intelligible fashion and written in standard English?

Reviewer #1: Yes

Reviewer #2: No

Reviewer #1: Dear Editor,

Dear Editor, I would like to express my gratitude to you and the Associate Editor for allowing me to review the manuscript “Risk and Protective Factors that distinguish those who have thoughts of self-harm or suicide from those who act on them: A Systematic Review in young people”. I approached the article with great enthusiasm and believe that it offers valuable insights into risk factors of suicidal thoughts of self-harm with a focus on critical thinking. The study's findings are quite revealing among young people. It indicates that risk factors that distinguished self-harm behavior from thoughts of self-harm included being female, exposure to self-harm/suicide, and impulsivity. However, to enhance the manuscript, the authors should have considered certain considerations, which would have made it more appealing to international readers.

General comment: Many typographical and grammatical errors should be attended to. Therefore, the writing needs language, grammar, and punctuation revision.

Regarding the specific aspects of the article:

Dear authors, Thank you for conducting such a valuable study. I have reviewed your manuscript thoroughly and come up with the following comments section by section.

Title: Risk and protective factors that distinguish those who have thoughts of self-harm or suicide from those who act on them: A systematic review in young people.

Abstract: Please clarify the abstract, particularly regarding the results. It is currently vague and lacks a concise summary, which may discourage researchers from reading it if published as is. Therefore, please rewrite the entire abstract.

Organize the abstract into the following sections according to the journal guidelines: background, methods, results, and conclusions.

Introduction

The literature review in the manuscript is comprehensive and timely. I recommend that the authors incorporate specific evidence and references from the literature to bolster their arguments, enhancing the credibility of their findings and offering readers a broader perspective on the topic.

Chosen methods, results, and data interpretation are within the higher-quality papers; however, they are missing originality: If there are plenty of studies according to your findings, why is your study unique? Several studies, including meta-analyses, have examined protective and risk factors associated with suicidal behaviors, such as thoughts, ideation, attempts, and completed suicide. Therefore, the authors need to justify the necessity of this study more effectively.

Research question

What risk and protective factors distinguish young people (aged 13-25 years) who have thoughts of self-harm or suicide from those who act on them? What age classification for young people (13-25 years) did the WHO establish?

How did you see the age category of children and adolescents according to WHO? This needs clear justification.

The methodology section lacks clarity on the criteria for including and screening literature. The authors should explicitly state these criteria and detail how they selected the articles for their review.

Please clarify your inclusion and screening criteria, as they are currently vague and not briefly summarized, which may deter researchers from reading your work if published in this manner.

Did you gather data on suicidal ideation and attempts over the past month, 12 months, or a lifetime, including the instruments used? Additionally, did you assess self-harm and suicidal ideation and behaviors together as a single outcome (suicidality)? Please clarify.

Results

Generally, your results part is not written as it fits scientific journals. Rewrite it, including descriptive parts.

Why did you exclude studies examining risk or protective factors for self-harm ideation and self-harm behavior or suicidal ideation and suicide attempts? And also, what about qualitative studies ?

What were the methodological quality assessment scores for studies on suicidal ideation and behaviors, which encompass all aspects from thoughts to actions? How did you handle data quality management?

Discussion

This review aimed to synthesize findings from existing research that have examined risk or protective factors in young people that distinguish between those who have thoughts of self-harm or suicide and those who act on their thoughts. Therefore, what was the outcome? either risk protective / both? If the outcome was both a risk and protective factor, address each separately.

Your discussion is too shallow, so discuss it in more depth.

Discuss your study with a systematic review and meta-analysis. Are there differences? If so, why? Compare your independent predictors with other studies. Discuss in more depth.

You compared your study to a single cross-sectional study in the discussion section. Is that appropriate? Please revise it.

Generally, your comparisons in the discussion part were shallow. So revise it and put your impression/insight based on your findings.

Thank you once again for the opportunity to review this work.

Reviewer #2: Dear Author,

Thank you for the opportunity to review this manuscript. Below are the comments on my review:

1. The title is no clearly reflects the subject matter of the article. It needs minor revision to make the object of study of the research clear.

2. In the abstract, clearly present your findings within each subsection.

3. In the Introduction section, please cite a reference for line 64. Additionally, this section should define self-harm, suicide attempt, and other suicidal behaviors such as suicide ideation.

4. Lines 71-75 describe a research gap and your current study's solution. Given that your study is descriptive and does not involve primary research or meta-analysis, the approach to addressing this gap should be clearly articulated. Furthermore, it is recommended to moving these sentences to the concluding section of the introduction.

5. Paragraph 2 line 77-87 needs revision and revises it based on scientific study

6. Superficial literature review: The introduction does not critically evaluate existing studies or highlight gaps in the literature, particularly in low- and middle-income countries.

• Lack of theoretical framework: The rationale for selecting specific independent variables (e.g., gender difference, and other mental illness that causes suicide Behaviors) is not well-explained.

7. For the limitations section, you must be more realistic.

Finally, it is a very important, informative, innovative, and insightful study filling the lacuna from your part of the world.

**Do you want your identity to be public for this peer review?** For information about this choice, including consent withdrawal, please see our Privacy Policy

Reviewer #1: **Yes: ** Tesfaye Segon

Department of Psychiatry, College of Medicine and Health Science, Injibara University, Injibara, Ethiopia. tesfayes721@gmail.com

Reviewer #2: No

---

## [Author Response · Author response to Decision Letter 1]

16 May 2025

Response to reviewer letter is attached.

---

## [Decision Letter · Decision Letter 1]

Exploring risk and protective factors which distinguish suicidal and self-harm behaviours from suicidal and self-harm ideation in young people: A systematic review.

PONE-D-24-41487R1

Dear Dr.Marianne,

We’re pleased to inform you that your manuscript has been judged scientifically suitable for publication and will be formally accepted for publication once it meets all outstanding technical requirements.

Kind regards,

Alemayehu Molla Wollie

Academic Editor

PLOS ONE

Additional Editor Comments (optional):

Reviewers' comments:

Reviewer's Responses to Questions

**Comments to the Author**

Reviewer #1: All comments have been addressed

Reviewer #2: All comments have been addressed

2. Is the manuscript technically sound, and do the data support the conclusions?

Reviewer #1: Yes

Reviewer #2: Yes

3. Has the statistical analysis been performed appropriately and rigorously?

Reviewer #1: Yes

Reviewer #2: No

4. Have the authors made all data underlying the findings in their manuscript fully available?

Reviewer #1: Yes

Reviewer #2: Yes

5. Is the manuscript presented in an intelligible fashion and written in standard English?

Reviewer #1: Yes

Reviewer #2: Yes

Reviewer #1: All the comments were addressed.

Dear Editor,

I have reviewed the manuscript and would like to provide specific comments for your consideration.

1. The literature review in the manuscript is comprehensive and timely. However, I suggest that the authors include more specific evidence and references from the literature to support their arguments. This will strengthen the credibility of their findings and provide readers with a broader perspective on the topic.

Reviewer #2: all required questions have been answered by Authors and that all responses meet formatting specifications

**Do you want your identity to be public for this peer review?** For information about this choice, including consent withdrawal, please see our Privacy Policy

Reviewer #1: **Yes: ** Tesfaye Segon.

Department of Psychiatry, College of Medicine and Health Science, Injibara University, Injibara, Ethiopia.

Tel: +251933214662.

Email address: tesfayes721@gmail.com.

Reviewer #2: **Yes: ** Biazin Yenealem

---

## [Editor Report · Acceptance letter]

PONE-D-24-41487R1

PLOS ONE

Dear Dr. Etherson,

I'm pleased to inform you that your manuscript has been deemed suitable for publication in PLOS ONE. Congratulations! Your manuscript is now being handed over to our production team.

Kind regards,

on behalf of

Mr. Alemayehu Molla Wollie

Academic Editor

PLOS ONE